# Cr-spinel records metasomatism not petrogenesis of mantle rocks

Hamed Gamal El Dien [1,2]*, Shoji Arai[3], Luc-Serge Doucet[1], Zheng-Xiang Li [1], Youngwoo Kil[4], Denis Fougerouse [5,6], Steven M. Reddy [5,6], David W. Saxey [6] & Mohamed Hamdy [2]

Mantle melts provide a window on processes related to global plate tectonics. The composition of chromian spinel (Cr-spinel) from mafic-ultramafic rocks has been widely used for tracing the geotectonic environments, the degree of mantle melting and the rate of mid-ocean ridge spreading. The assumption is that Cr-spinel's core composition ($Cr\# = Cr/(Cr + Al)$) is homogenous, insensitive to post-formation modification and therefore a robust petrogenetic indicator. However, we demonstrate that the composition of Cr-spinel can be modified by fluid/melt-rock interactions in both sub-arc and sub-mid oceanic mantle. Metasomatism can produce Al-Cr heterogeneity in Cr-spinel that lowers the Cr/Al ratio, and therefore modifies the Cr#, making Cr# ineffective as a geotectonic and mantle melting indicator. Our analysis also demonstrates that Cr-spinel is a potential sink for fluid-mobile elements, especially in subduction zone environments. The heterogeneity of Cr# in Cr-spinel can, therefore, be used as an excellent tracer for metasomatic processes.

[1] Earth Dynamics Research Group, The Institute for Geoscience Research (TIGeR), School of Earth and Planetary Sciences, Curtin University, GPO Box U1987Perth, WA 6845, Australia. [2] Geology Department, Faculty of Science, Tanta University, 31527 Tanta, Egypt. [3] Department of Earth Science, School of Natural System, Kanazawa University, Kanazawa 920-1192, Japan. [4] Department of Energy and Resources Engineering, College of Engineering, Chonnam National University, Yongbong-ro, Buk-gu, Gwangju, South Korea. [5] School of Earth and Planetary Sciences, The Institute for Geoscience Research (TIGeR), Curtin University, GPO Box U1987Perth, WA 6845, Australia. [6] Geoscience Atom Probe, Advanced Resource Characterisation Facility, John de Laeter Centre, Curtin University, GPO Box U1987Perth, WA 6845, Australia. *email: hamed.gamaleldien@postgrad.curtin.edu.au

Mantle partial melting is a fundamental process that has contributed to the chemical stratification of the Earth and plays a key role in the long-term evolution of the lithosphere and Earth's tectonic and geodynamic processes in general[1]. Irvine[2,3] first proposed to use chromian spinel ((Mg, Fe$^{+2}$) (Cr, A1, Fe$^{+3}$)$_2$O$_4$: hereafter Cr-spinel) in mafic-ultramafic rocks as a petrogenetic indicator because, at magmatic temperatures, this mineral is highly sensitive to the chemical conditions associated with melt generation in the mantle. Numerous subsequent studies have used the composition of Cr-spinel to estimate the degree of melt extraction, and therefore to discriminate among the different tectonic settings of ophiolitic complexes[4]. Arai[5] systematically assessed spinel peridotites in terms of mantle melting conditions mainly based on the Cr# (Cr/(Cr + Al)) of Cr-spinel in combination with olivine composition. Cr-spinel has thus been established as a reliable indicator for studying mantle petrology[6], where the Cr# of Cr-spinel in residual peridotites has been viewed as a quantitative melting indicator for mantle residue based on the positive correlation between Cr# in Cr-spinel and degree of melting (see Fig. 3 of Hellebrand et al.[6]).

However, mantle peridotites generally experienced complex metasomatic and metamorphic processes that can potentially obliterate their primary residual origin. Cr-spinel can exhibit a chemical classic (normal) zoning due to melting and fractional crystallization processes characterized by Mg-Al-rich core and Cr-Fe$^{+2}$-rich rim[4], or metamorphism that commonly removes almost all the Al from the outer parts of the crystal, leaving behind magnetite/ferritchromite rims[7]. Moreover, reverse zoning (Mg- and Al-rich rim and Fe$^{+2}$- and Cr-rich core) can also occur, and it is usually regarded as a result of stress and deformation[8], elemental exchange with co-existing silicates[9], or melt/rock interaction[10]. Reverse zoning is undetectable under an optical microscope, and it appears either concentric and/or asymmetric when observed in atomic number contrast under backscattered electron (BSE) imaging[9], or remains non-observable under both an optical microscope and BSE (this study).

Here, we conduct a correlative microanalysis workflow from the grain-scale to the nano-scale using X-ray elemental mapping, electron microprobe (EMP), laser-ablation-inductively coupled plasma mass spectrometry (LA-ICPMS), electron backscattered diffraction (EBSD), and atom probe tomography (APT) in order to describe the Cr-spinel reverse zonation in mantle peridotites from the Arabian Nubian Shield. This type of reverse zoning "Al-Cr heterogeneity" is characterized by Al enrichment and Cr depletion in the rim compared to the core of the studied crystals. We demonstrate that this reverse zoning is unrelated to the magmatic or metamorphic/deformation history but is due to fluid/melt-rock interactions (i.e., metasomatism). The most striking feature is that such Cr-spinel Al-Cr heterogeneity is a widespread feature that affects the mantle rocks of various tectonic settings and ages (Supplementary Fig. 1 and Supplementary Data 1). We therefore question the robustness of Cr-spinel during post-formation modification, and thus the suitability of Cr-spinel as a straightforward reliable indicator for tracing mantle melting conditions and tectonic environments.

## Results

### Al-Cr reverse zoning in spinel: An example from the Arabian-Nubian Shield.
The Arabian-Nubian Shield (ANS) represents the largest Neoproterozoic juvenile continental crust formed through accretion of island arcs to continental margins by the closure of the Mozambique Ocean during the East-African orogeny (750–550 Ma)[11,12]. The peridotites in this study were sampled from serpentinite bodies at Wadi Alam in the Central Eastern

Desert of Egypt (Supplementary Fig. 2). Petrographic and textural investigations show that our peridotite samples consist of serpentine minerals, olivine relics, orthopyroxene bastites, carbonates, and Cr-spinels, with no amphiboles or chlorite. The samples have been affected by varying degrees of serpentinization. The predominance of pseudomorphic textures as mesh and bastite reflects a harzburgite protolith (Supplementary Fig. 2). For more details about the geological background of the ANS see Gamal El Dien et al.[13], and for field observations and petrographical and mineralogical descriptions of the studied rocks see Hamdy and Gamal El Dien[14].

The petrological and geochemical characteristics of the studied peridotites reveal a highly depleted origin, as shown by (1) a harzburgitic, clinopyroxene-free modal composition, (2) their low Al$_2$O$_3$ and high MgO bulk-rock content (<1 wt% and >44 wt%, respectively) (Supplementary Fig. 3 and Supplementary Data 2) and (3) their low, heavy rare earth elements content (HREE; Yb$_{N(CI-normalized\ value)}$ = 0.01–0.04) and low Y (<0.10 ppm) (Supplementary Fig. 4). Melting models, using non-modal fractional melting, reproduced the HREE values of our samples with a 25–30% melt extraction from a depleted MORB mantle (DMM) source[15,16] (Supplementary Figs. 4, 5). This supports the highly refractory origin for the studied mantle peridotites. The bulk-rock major and trace element contents, and REE patterns, are similar to mantle wedge peridotites[17–20] (Supplementary Figs. 3, 4). Despite their refractory origin, the studied peridotites exhibit enrichment in incompatible trace elements and light REE (LREE) (Supplementary Fig. 4). These enrichments are not correlated with either serpentinization (i.e., loss on ignition) (Supplementary Fig. 6) or the different melting indices used (Supplementary Fig. 3i), which indicates that the rocks underwent a post-melting metasomatic process[21]. This metasomatism process is believed to be due to fluid/melt-rock interaction between mantle wedge peridotites and slab-derived melts, as supported by a high enrichment of high field strengths elements (HFSE) (Supplementary Fig. 7) and fluid-mobile elements (FME) in those rocks[14]. Generally, migration of aqueous fluids/melts from a subducting slab enriches the overlying mantle with incompatible elements (i.e., FME) and water, yielding metasomatized sub-arc mantle[22,23].

Cr-spinel represents up to 2% of the mineral modal composition in our samples and exhibits a holly-leaf shape with a red color characteristic of residual peridotites[24]. It presents as anhedral to subhedral grains of 50 μm to 2 mm in size. In BSE images, all Cr-spinel grains display small magnetite rims either surrounding a homogenous core and/or appearing in open cracks, indicating a late-stage modification (i.e., serpentinization and/or metamorphism; Fig. 1a and Supplementary Fig. 8). However, X-ray elemental mapping shows strongly heterogeneous and modified cores (Fig. 1b–f). These cores display a concentric and gradual (Fig. 1b, c, e, f) or asymmetrical (Fig. 1d) reverse zoning of highly Al-enriched and Cr-depleted rims and the opposite in the cores (we called this heterogeneity hereafter) with less variation in the distribution of Mg and Fe$^{+2}$ (Fig. 2, Supplementary Fig. 8, and Supplementary Data 2).

The cores of Cr-spinels show a wide variation in compositions between grains from the same sample (core-to-core: Figs. 1c, f and 2f), among samples (Figs. 1c–f and 2c–e), and within the same grain (core to rim: Figs. 1c–f, 2f and Supplementary Data 2). The Al$_2$O$_3$ content and Cr# display a continuous reverse variation trend from the cores (14.6–18.3 wt%, and 0.64–0.70, respectively) to the rims (19.6–26.1 wt % and 0.53–0.62, respectively) (Figs. 1c–f and 2c–f). Al$_2$O$_3$ content shows different covariation trends, from the cores to the rims, with transition elements such as Sc (negative correlation), Ti and Ni (positive correlation), Cu and Ga (no correlation), whereas V, Mn, Co, and Zn show V-shape

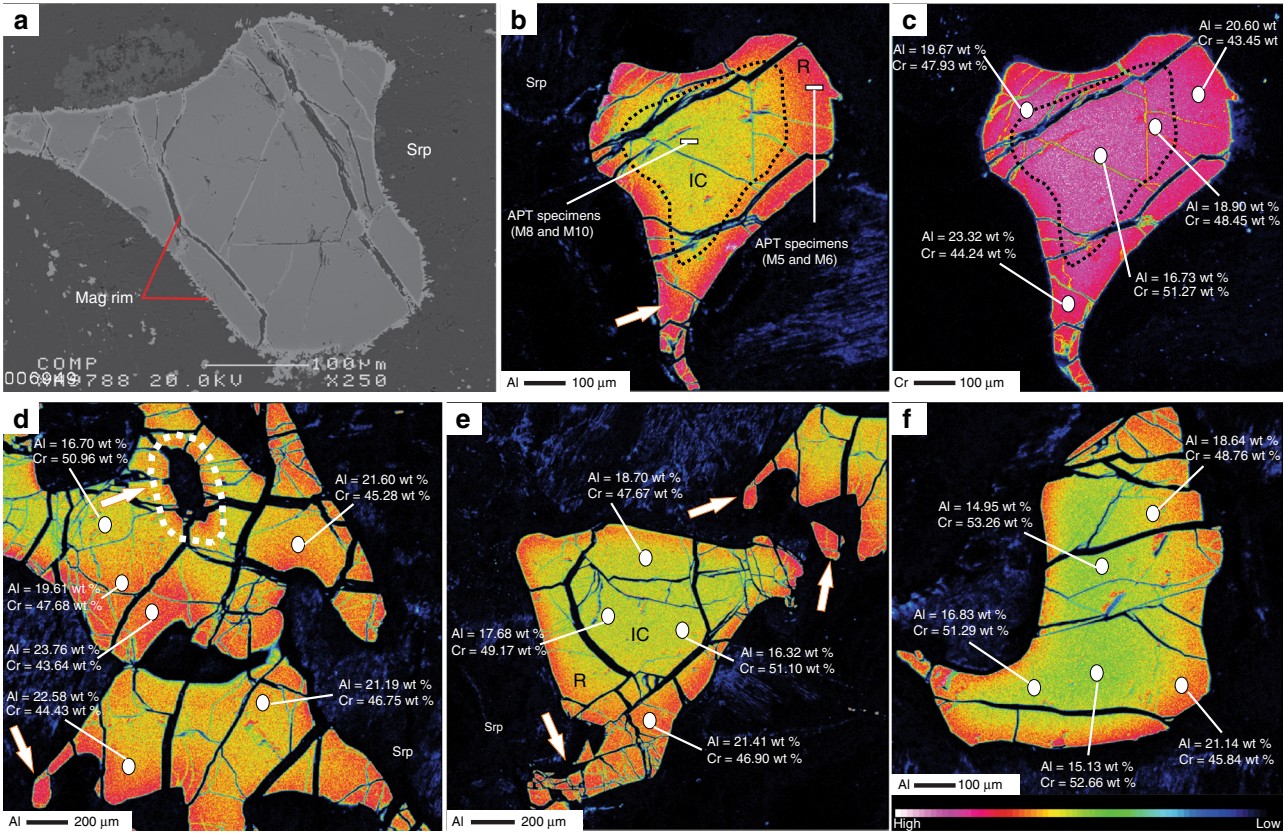

**Fig. 1** X-ray elemental map of the studied Cr-Spinel. **a** Backscattered electron (BSE) image of a Cr-spinel grain with a homogenous core surrounded by a small magnetite rim. Al (**b**) and Cr (**c**) X-ray map for the same grain show reverse zoning for Al and Cr. Asymmetrical and heterogeneous distribution of Al-Cr within the core (**d**) and gradual increase of Al-Cr from core to rim (**b**, **c**, **e**, **f**) of three different Cr-spinel grains. White arrows point to Al halos around inclusions within the Cr-spinel grains (**d**) and high Al content in small grains with tightly curved rims (**b–e**). The results of the Atom probe tomography (APT) specimens for the core (M8 and M10) and rim (M5 and M6) are shown in Fig. 5. Mag = magnetite, Srp = serpentine, IC = inner core and R = rim

covariation trends (Supplementary Fig. 9). Fluid mobile elements (FME: Li, Rb, Sr, Cs, and Sb) increase with increasing $Al_2O_3$ content from cores to rims (Fig. 3). The opposite correlation is found between those elements and $Cr_2O_3$ content. We thus used Al content as a representative for these elemental variations in the following discussion.

Metasomatism is a ubiquitous phenomenon in the Earth's mantle and can be a protracted chemical process that modifies the primary chemical composition of pre-existing rocks and their composing minerals[25]. Metasomatism can be identified by drastic changes in the rocks' mineralogy (called modal metasomatism or refertilization) or by subtly incompatible trace element enrichment in the rocks and minerals, also called cryptic metasomatism[26]. This process can happen in a sub-arc mantle (for example, arc-peridotites[27,28]), and in a sub-oceanic mantle (i.e., Mid Ocean Ridge (MOR)-peridotites[29]).

The most striking feature of the studied peridotites is the positive covariation between Al content and FME (Li, Rb, Sr, and Cs) contents in Cr-spinel (Fig. 3a–d). This shows that slab-derived fluids-rock interactions were responsible for the Al-Cr heterogeneities (i.e., addition/depletion) in Cr-spinel, rather than fractional crystallization/melting[4,30], subsolidus elemental exchange[9], or stress[8]. The primitive mantle-normalized pattern of the FME of the studied Cr-spinel (Fig. 3e) shows similarities to the average pattern of subduction inputs which includes altered oceanic crust (AOC)[31], global subducted sediments (GLOSS II)[32], and marine sediments[33]. The studied Cr-spinels also have similar FME contents as melt inclusions in Cr-spinels from the Avacha peridotite xenoliths, which experienced interactions with slab-

derived melts[34] (Fig. 3e). Moreover, FME in the studied Cr-spinel cores and rims are highly enriched relatively to primitive mantle[35], in contrast to Cr-spinel in refractory/depleted peridotites[36] (Fig. 3e). This indicates that the compositions of both their cores and rims have been modified. We interpret the Al, Cr and FME zoning in Cr-spinel to be the result of cryptic metasomatism by interactions of hydrous Al-rich slab-derived melts with the host peridotites.

Deformation/stress can led to Al and Cr crystal lattice diffusion resulting in Al-Cr dipolar zoning in elongated Cr-spinel grains in deformed peridotites[8]. Contrary, the studied peridotites have massive textures at the field scale with no foliation/schistosity (Supplementary Fig. 2c) and minerals aggregates have rounded shape and granular texture without any elongation and/or lineation arrangement (Fig. 1 and Supplementary Fig. 2d–h). In contrast to the dipolar zoning, the studied Cr-spinels show a concentric Al-Cr zonation (Fig. 1 and Supplementary Fig. 8). To test the effect of deformation/stress and microstructure of Cr-spinel grains, EBSD has been used to quantify the crystallographic orientation and microstructural characteristics of some Cr-spinel grains (Fig. 4). The maximum misorientation in each grain from these points is 10° and 15° for grain 1 and 2, respectively (Fig. 4a, b). This misorientation variation is spatially linked to the late brittle fractures that cut the grains (including the Al-Cr zoning seen in the X-ray elemental maps data (Fig. 1b, f)). Within individual, fracture-bound fragments of the grains, there is no evidence for any lattice orientation variation or significant plastic deformation. This includes fragments that contain the observed core to rim compositional variations. Hence, there is no

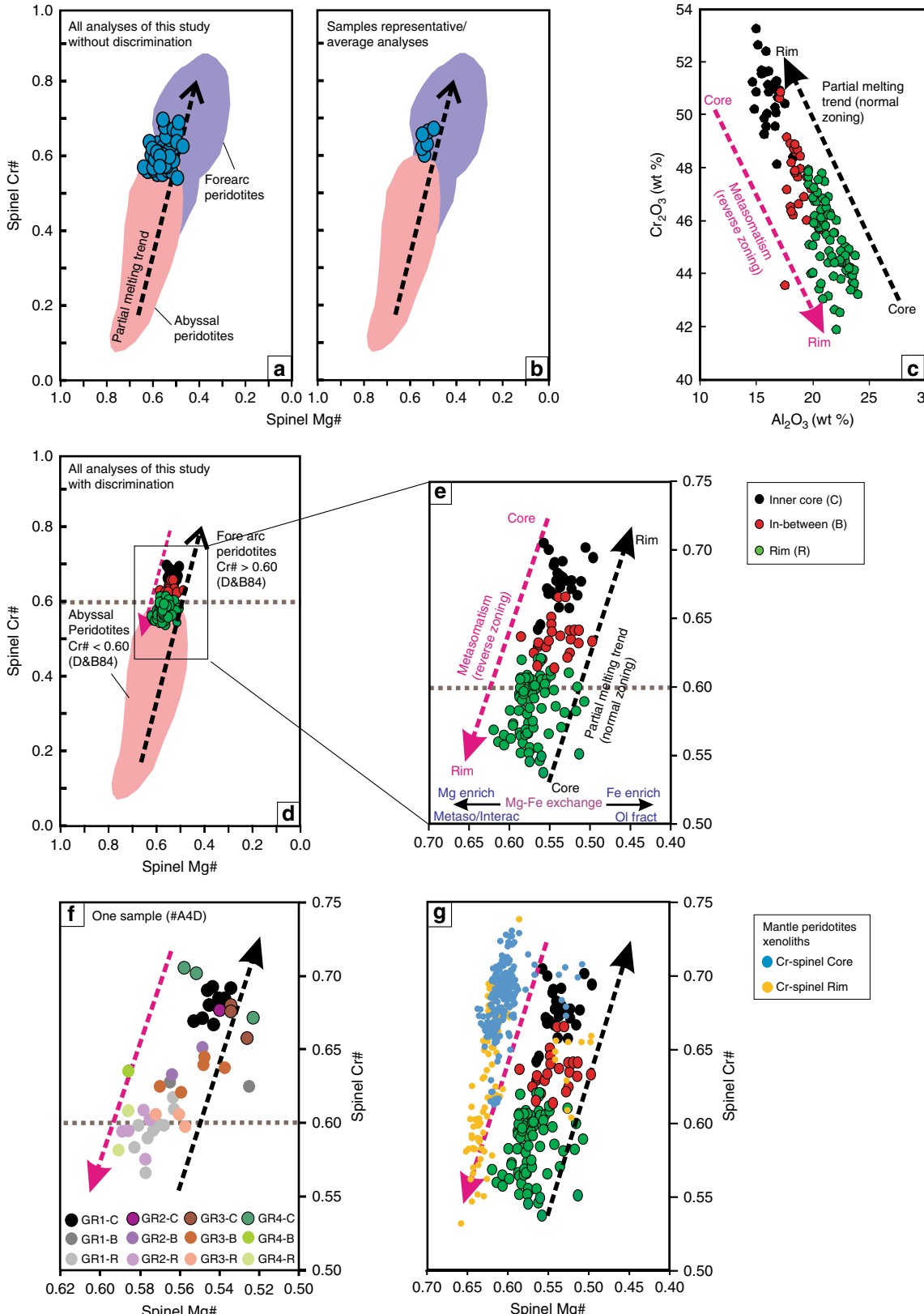

relationship between the observed compositional Al-Cr heterogeneity in Cr-spinel and grain-scale deformation. In addition, the $Fe^{3+}/(Fe^{3+} + Fe^{2+})$ ratio, which is widely use as metamorphism-related enrichment indicator of the Cr-spinel[7], has a constant values between the Cr-spinel cores and rims and has no relationship with Al and FME (Supplementary Fig. 10) giving a

clear evidence that the enrichment process of those elements is not related to the late stage metamorphic processes.

On the crystal scale of the studied Cr-spinel, the interpreted metasomatism process is demonstrated by high Al content in fine-grained Cr-spinel and in tightly curved rims that would have been more affected by melt/rock interaction than coarser

**Fig. 2** Plots of the studied Cr-Spinel chemical composition for different zones within the grains. **a** Cr# (Cr/Cr + Al) vs. Mg# (Mg/Mg + Fe$^{+2}$) plot of all the raw datasets for Cr-spinel grains in the studied rocks. Such data are usually not published in previous literature. The data span between recommended fields used in previous literature for abyssal peridotites[4] and fore arc (FA)-peridotites[40,42]. **b** Representative/average Cr-spinel data for each sample. Such sample-average data are usually used in previous mantle petrology studies. This plot shows that our samples had a fore arc to abyssal peridotite origin. **c** Al$_2$O$_3$ vs Cr$_2$O$_3$ and **d** Cr# vs Mg# plots of the studied Cr-spinel grains with low Al$_2$O$_3$ and high Cr$_2$O$_3$ and Cr# in their cores, and high Al$_2$O$_3$ and Low Cr$_2$O$_3$ and Cr# in their rims (reverse zoning). Partial melting trend and the brown line between abyssal peridotites (Cr# < 0.60) and FA-peridotites (Cr# > 0.60) are from Dick and Bullen[4]. **e** All datasets have trends parallel to both the melting trend and our newly defined metasomatism trend. **f** Datasets of different zones (from core to rim) from four grains from a single sample (sample # A4D) that show large Al-Cr heterogeneity. The data span across the whole range between abyssal peridotites and FA- peridotites. **g** Plots of the studied Cr-spinel grains compared with the compositions of both modified spinel/rims and non-modified spinel/cores of published mantle peridotite xenoliths[17,18,34,39]

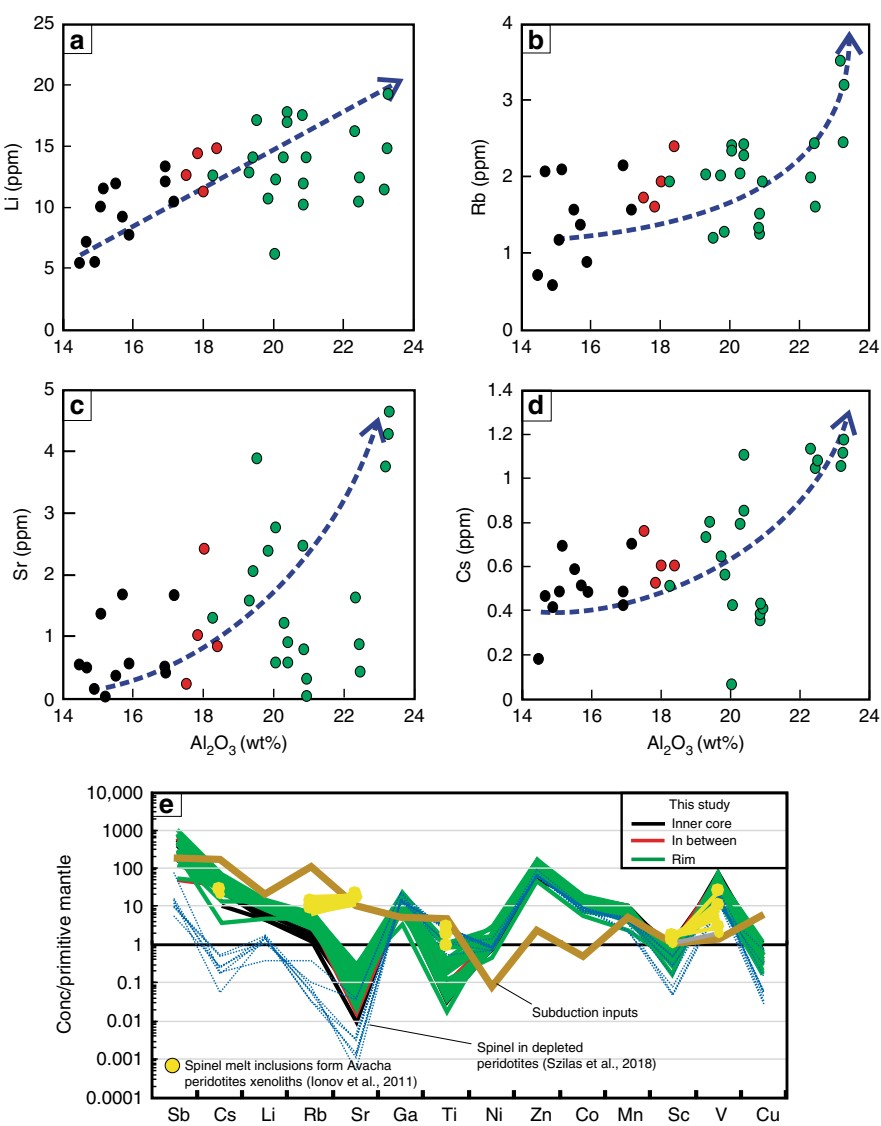

**Fig. 3** Trace elements concentration of different zones in the studied Cr-Spinel. **a–d** Covariation between Al$_2$O$_3$ (wt %) content and fluid-mobile elements (FME: Li, Rb, Sr, and Cs). All the elements show positive a correlation with Al content and an increase from core to rim. **e** FME and transition elements normalized to primitive mantle[35] compare with the average content of subduction inputs including altered oceanic crust (AOC)[31], global subducted sediments (GLOSS II)[32] and marine sediments[33], melt inclusions in Cr-spinel from Avacha peridotite xenoliths[34], and spinel in refractory/depleted peridotites[36]. The Cr-spinel show high enrichment in FME, attributed to slab-derived fluid/melt interaction with host peridotites

grains[17,19,37] (Fig. 1b–f); idiomorphic shapes for some grains (Fig. 1b) that reflect high degrees of melt/rock interaction, where melt diffusion into the peridotites not only modified the composition of the Cr-spinel, but also corroded and modified the crystal morphology;[38] and Al-rich halos around inclusions, which is expected to form during melt/rock interaction inside Cr-

spinel grains, whereas the trapped melts modified the surrounding Cr-spinel (Fig. 1d and Supplementary Fig. 8c) similar to high Al zone reported around melt inclusions in Cr-spinel from the Avacha peridotite xenoliths[34,39].

The observation that some grains have Al-Cr heterogeneity extended inside the grain up to 200 μm (Fig. 1) suggests that such

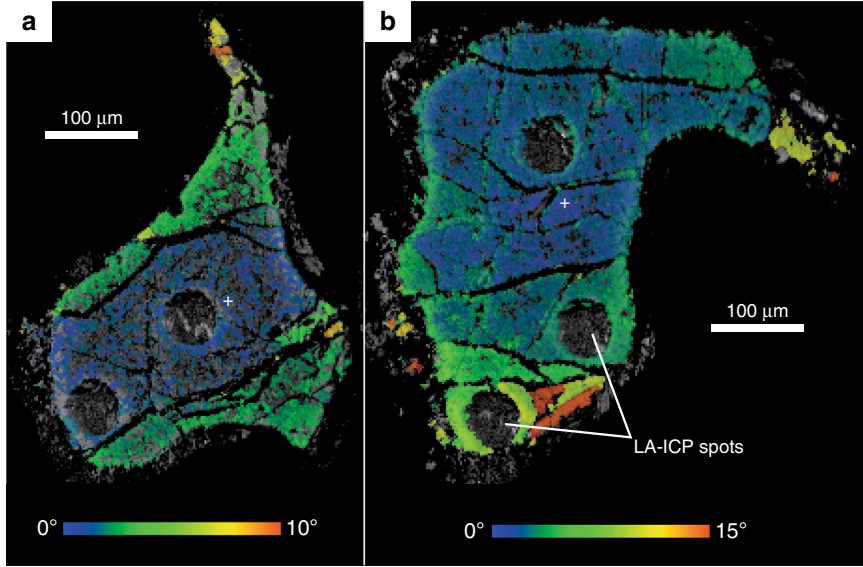

**Fig. 4** EBSD microstructural data from two Cr-spinel grains from sample A4D. Images comprise greyscale image of EBSD pattern quality overlain by misorientation maps measured relative to the orientation of spinel lattice at the position shown by the white cross. For grain **a** total misorientation is 10°, for **b** the total misorientation is 15°. The change in misorientation in each grain corresponds to presence of late fractures seen in the pattern quality image. Total misorientation within individual fracture-bound regions of the grain is <1°, indicating that there is no plastic deformation within the grains

heterogeneity in the grains is not an artifact of a deep embayment of Cr-spinel grains during cutting, i.e., sectioning effect. In addition, LA-ICPMS data confirm the existence of such heterogeneity in different zones of the Cr-spinel (Fig. 3). To further verify the 3D Al-Cr heterogeneity in the studied Cr-spinel, we applied APT advanced technique, which is a powerful tool to characterize and construct the 3D chemistry at the nanoscale (see methods for more details). Four needle-shaped specimens (Fig. 5) from the core (M8 and M10) and the rim (M5 and M6) were extracted from sample A4D (Fig. 1b). The specimens yielded between 100 million atoms (M5, M6 and M10) and 109 million atoms (M8). The core specimens M8 and M10 have Cr = 24.1 atomic % (at%) and 24.4 at%, respectively and Al = 9.0 at% and 8.9 at%, respectively (Fig. 5 and Supplementary Data 2). On the other hand, the rim specimens M5 and M6 have Cr = 22.1 atomic % (at%) and 22.3 at%, respectively and Al = 10.7 at% and 10.6 at %, respectively (Fig. 5, Supplementary Data 2 and Supplementary Movies 1 and 2 for Al in M6 and M10 specimens, respectively). Although the major element composition calculated from the APT data differs from EMPA and LA-ICPMS data due to the lack of standardization protocols, the APT results confirm the enrichment of Al and depletion in Cr in the rim, and the opposite for the core (Fig. 5). Also, the detection limit of APT for the FME in Cr-spinel is too high to make meaningful measurements. The homogenous distribution of Al, Mg and Fe, non-detection of Si, and absence of any isolated clusters across the specimens, thus indicate that the high FME concentration in Cr-spinel is inherited and not related to silicate inclusions (i.e., serpentine phases), Fe-oxide nano-scale inclusions (magnetite) or low temperature alteration (Fig. 5).

The above observations make the measured Cr-spinel composition unsuitable for deciphering the partial melting history of the studied peridotites. The melt/rock interaction between mantle peridotites and slab-derived fluids/melts may produce a strong heterogeneity that modifies the Al and Cr contents (i.e., Cr#) of primary Cr-spinel, and therefore produces a reverse trend/range (metasomatism trend) of Cr# in the studied mantle rocks, different from the melting trend. Plotted together on the Cr# vs. Mg# diagram (Fig. 2d), the Cr# of the rims (mostly <0.6) are similar to MOR-peridotites, and the cores (with Cr# > 0.60) are

similar to Fore arc (FA)-peridotites[4]. This indicates that representative and/or average Cr# values of Cr-spinel could give a misleading conclusion about the geotectonic setting of our samples (Fig. 2a, b). For example, applying the equation of Hellebrand et al.[6] $[F = 10\ ln\ (Cr\#) + 24]$ to the studied Cr-spinel would indicate 17–18% melting if using the values from the rim, and >20% melting when using the values from the core. Such values are not consistent with the bulk rock data (Supplementary Figs. 4, 5). We therefore conclude that for peridotites that experienced post-melting fluid/melt-rock interactions, their Cr-spinel data should not be used for determining the tectonic setting and melting history.

**A re-evaluation of Al-Cr heterogeneity in Cr-spinel from previous studies.** Our study shows that metasomatism can dramatically change the composition of Cr-spinel (i.e., Cr#). A careful literature review revealed that Cr-spinel compositional heterogeneity can be found in most mantle rocks of various tectonic environments and ages (Supplementary Fig. 1 and Supplementary Data 1). These include all available Cr-spinel data from arc-peridotites composed of FA-peridotites (including dredged samples from the present-day oceanic arc, e.g., the Izu-Bonin-Mariana arc, and mantle wedge xenoliths such as those from the Kamchatka arc) and back-arc peridotites (i.e., Mariana Trough), plus abyssal/MOR-peridotites (Supplementary Data 3).

Strikingly, even though some of the reported Cr-spinel have homogeneous compositions, the majority of the reported Cr-spinel from arc-peridotites, FA-peridotites in particular, show high heterogeneity in Al and Cr contents both within single grains (with core-rim structures)[17,40,41] and within a single sample (from core to core)[42–44], which we hereafter refer to as modified Cr-spinel. Based on the available data from arc-peridotites (Fig. 6a), we filtered the modified/non-modified Cr-spinel using the Al-Cr variability between the core and rim of a single grain, and between cores of grains from the same sample (see methods). The non-modified Cr-spinel grains that have tight Cr# for a single sample display a different Cr# range of ~0.45–0.70 (Fig. 6d) for FA-peridotites from that of the accepted range used in previous studies (~0.30–0.85) (Fig. 6a), and back

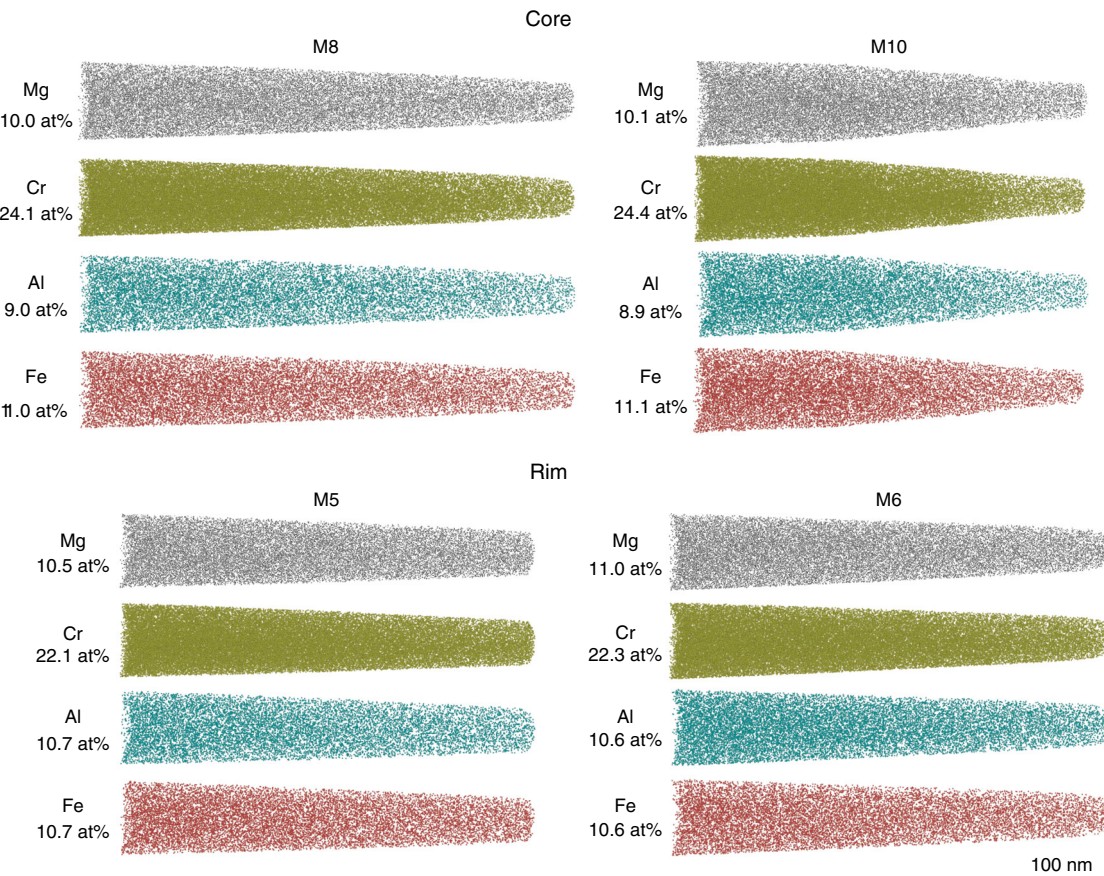

**Fig. 5** Atom probe tomography results. Atom maps of Mg, Cr, Al, and Fe are presented for needle-shaped specimens from the core (M8 and M10) and the rim (M5 and M6). The specimens were extracted from sample A4D and their location is indicated on Fig. 1b. The composition in atomic % is indicated for each specimen (See Supplementary Data 2). The composition of the core is enriched in Cr and depleted in Al compared to the rim

arc-peridotites have Cr# range of ~0.15–0.65 that significantly overlapped with FA-peridotites (Fig. 6a, c).

In contrast, metasomatism is slightly different in abyssal/MOR-peridotites due to the rarity of hydrous fluids. It is driven by a melt-rock interaction that causes veined peridotites[29], LREE enrichment in clinopyroxene[45,46], and Ti enrichment in Cr-spinel[29,47]. MOR-peridotites can be divided into two main types residual rocks (not affected by any melt-rock interaction process) and non-residual rocks (including dunites, plagioclase peridotites, gabbroic-pyroxenite veined samples, and metasomatized ones)[29]. However, both rock types share the same range of spinel Cr# (~0.1–0.6) (Fig. 6b) following the melting trend[4] (Supplementary Fig. 11) and the non-residual rocks are present in all the ridges. This indicates that the Cr-spinel from sub-oceanic mantle beneath all ridge systems is likely to have been affected by similar melt-rock interaction processes.

Representative Cr# of Cr-spinel composition for a given location or dredge base has been used for mitigating the effect of Cr-spinel heterogeneity and high compositional variations[29,48]. However, this approach overlooked the elemental compositional variability of the Cr-spinel affected by melt-rock interaction. To further illustrate this point, we discuss below and give specific examples, according to available published datasets, of Cr-spinel heterogeneity due to metasomatism at different scales, from dredge sites to samples (core-core Al-Cr heterogeneity) and single grains (within core and core-rim Al-Cr heterogeneity) (see Supplementary Data 1 for more details and summary of Cr-spinel heterogeneity for each individual ridge at different scales, and Supplementary Data 3 for the complete datasets). Here we emphasis that using representative or average data for Cr-spinel

made it very difficult (or even impossible) to investigate the variation in Cr-spinel of those rocks. Hence, we only considered the samples that have at least two grains analyzed (see Supplementary Data 3). In addition, there have been only rare cases where studies traced core to rim chemical variation within individual grains[47,49–51].

There is a large variation in Al and Cr contents in Cr-spinel from a given dredge site, for example, section #V3306-IN18 (Owen FZ, Central Indian Ridge – CIR) has Al content = 32.20–52.90 wt% and Cr# = 0.15–0.42;[52] section #ANTP-89-HD (Marie Celeste TF-CIR: Al = 28.4–46.9 wt% and Cr# = 0.23–0.50);[53] section #PS55–89 (Lena trough- Arctic Ridge: Al = 25–55 wt% and Cr# = 0.14–0.54);[54] section # S1905 (Vema TF- Mid Atlantic Ridge, MAR: Al = 27.1–48.2 wt% and Cr# = 0.22–0.49)[49–51] and section #Van7–85 (Oblique Segment, South Western Indian Ridge – SWIR) has Al content = 24.41–54.87 wt% and Cr# = 0.16−0.52[55]. For more examples and details, see Supplementary Data 1.

Similar ranges of Cr# variation exist at the sample scale. For example, Hamlyn and Bonatti[52] analyzed four grains from the same peridotite sample (#V3306-IN18I-Owen FZ, CIR) and found inter-grain variations in Cr# spreading the range of 0.2 to 0.4 (mainly the entire range for CIR residual peridotites). Also, sample # AII32-8-6 (residual harzburgite-Ridge at 43°N, MAR: Al = 26.8–38.1 wt% and Cr# = 0.34–0.52)[56], sample # 1274A-14R-1/76-82 (residual harzburgite-Fifteen-Twenty TF, MAR: Al = 32.4–41 wt% and Cr# = 0.32–0.44)[57], sample #G9604-4 (residual peridotite-Conrad FZ, American Antarctic Ridge (AAR): Al = 23–40 wt% and Cr# = 0.29–0.53)[58] and sample #895D-4R2-45-49 (Hess Deep, East Pacific Rise (EPR): Al = 17.5–25.2 wt% and Cr# = 0.51–0.58)[59]. Dick et al.[47] published a

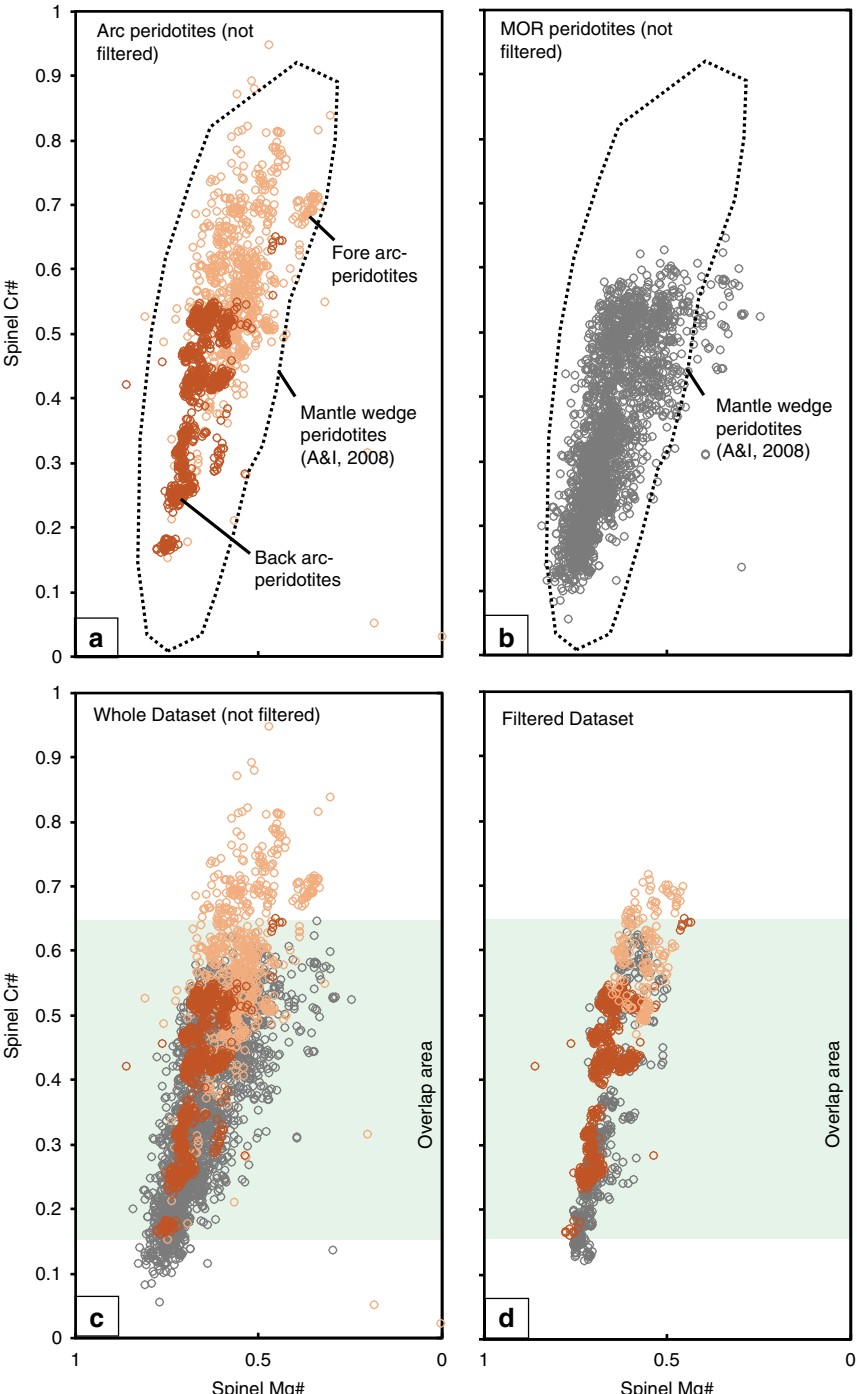

**Fig. 6** Spinel Cr# and Mg# for the fore arc peridotites. **a** Cr-spinel database from arc-peridotites including fore arc settings (composed of mantle wedge xenoliths (e.g., the Kamchatka arc)[18,19] and dredged samples form present-day oceanic arcs (e.g., the Izu-Bonin-Mariana arc)[40,42]) and back arc-peridotites (i.e., Mariana Trough)[43,44]. **b** Cr-spinel database from abyssal/ Mid-ocean ridge (MOR) peridotites. See Supplementary Data 3 for the arc and MOR-Cr-spinel datasets. All the arc and MOR-peridotites dataset plot within the Cr-spinel form mantle wedge peridotites[28]. **c** All arc peridotite Cr-spinel datasets (without filtering) plotted against MOR-peridotite one, showing their overlapping fields. **d** Filtered dataset for both arc- and MOR-peridotite are still completely overlapping. For filter details, see the methods. The shaded area in (**c**, **d**) marks the overlapping range of Cr# (0.15–0.65) between the arc-peridotite and MOR-peridotite fields

Cr-spinel dataset from residual peridotites from Kane Mega-mullion (MAR), typically with data of one grain only per samples except for sample Kn180-2-4-2 that has two grains with a large range of Al content = 40.0–36.5 wt% and Cr# = 0.30–0.36 (Supplementary Data 3). In the Warren[55] study, there are also a variation of Al-Cr content within the same sample of cryptic metasomatized peridotite from the Atlantis II fracture

zone (SWIR), such as sample #6K-465-2 that has a Al range of 28.89–46.54 wt% and Cr# of 0.22–0.46, and sample #RC27-9-6-5 that has a Al range of 27.59–46.58 wt% and Cr# of 0.22–0.48. For more examples and details, see Supplementary Data 1.

At the grain scale, data reported in Brunelli et al.[49], Cipriani et al.[50,51] and Dick et al.[47] have some grains showing within-grain

core-rim Al-Cr reverse variation. For example, sample #Kn180-214-44 (residual peridotite- Kane Megamullion) has rim showing high Al = 39.5 wt% and core showing low Al = 36.8 wt%[47], sample # L2627-04B (residual peridotite- Vema TF) has rim showing high Al = 48.5 wt% and core showing low Al = 44.5 wt%[51], sample # L2630-01D (residual peridotite- Vema TF) has rim showing high Al = 32.2 wt% and core showing low Al = 27.8 wt%[51]. For more examples and details, see Supplementary Data 1.

These examples, together with the comprehensive compilation (Supplementary Data 1), show that the majority (~85%) of Cr-spinel from MOR-peridotites exhibit heterogeneity, indicating that it is a widespread feature in sub-oceanic mantle (Supplementary Fig. 1 and Supplementary Data 1). Moreover, such large variations in Cr# featuring Al enrichment and Cr depletion are most likely the results of melt-rock interaction/metasomatism, consistent with cryptic metasomatic that led to LREE enrichments in Cpx of the same rocks (e.g., refs. [46,47,53]). For example, Warren[55] reported a similarly large compositional variation in Cpx from MOR-peridotites in the scales of dredge sites, samples, and individual grains, and interpreted it as the result of widespread melt-rock interaction. Indeed, some non-residual/impregnated samples have high Al content and low Cr# than residual samples from the same section, for example, sections # 6K-458 and Kn162-19 from SWIR (Supplementary Data 3)[55]. Moreover, Constantin et al.[60] documented that the Terevaka TF cryptic harzburgite samples have lower Cr# (~0.35) compared to harzburgites free of impregnation (Cr# = ~0.45; see their Fig. 2). Additionally, the orogenic replacive dunites (melt-rock interaction origin) have low Cr# than the residual origin dunites (see Fig. 8 of Su et al.[61]). Our re-interpretation of high Al-Cr heterogeneity in Cr-spinel from MOR-peridotites as a melt-rock interaction/metasomatism origin is supported the new geochemical modeling and observations of Brunelli et al.[62] from Vema at MAR where veined peridotites have lower Cr# than residual/vein-free ones.

We distinguished modified from non-modified Cr-spinel in MOR-peridotites, as we did for arc-peridotites, using our filter criteria (see methods). The data show a large overlap in Cr# of Cr-spinel between MOR-peridotites and arc-peridotites (~75% of all FA-peridotite data, ~90% of all MOR-peridotite data and 100% of back arc-peridotites data fall inside the common field of Cr# = ~0.15–0.65) (Fig. 6c, d). Furthermore, Arai and co-workers[28,63,64] suggested that the lithospheric mantle beneath arcs (such as the Western Pacific arcs) have peridotites with a chemical and mineralogical composition comparable to that of MOR/abyssal peridotites. In addition, Cr-spinel datasets of both modified and non-modified MOR-peridotites fall entirely within the field of mantle wedge peridotites[28] (Fig. 6b), rendering the application of Cr# as a geotectonic indicator ineffective, and Cr# < 0.6 is not a unique feature of MOR-peridotites.

## Discussion

Our new analyses and a careful review of published results show that the chemical composition of Cr-spinel can be easily and severely modified by cryptic metasomatism through fluid/melt-rock interaction involving slab-derived fluids in the sub-arc mantle, or even by small volumes of melts in the sub-oceanic mantle. We also demonstrate that the wide range of Cr# of Cr-spinel in mantle rocks is a function of melt-rock interaction[10,65–68] rather than partial melting. In addition, we identify Cr-spinel as a carrier of FME especially in subduction zone environments, where Al is easily mobilized by fluids and melts and therefore cannot be used as a melting degree indicator at least at the mineral scale. Furthermore, the heterogeneity of Cr# in Cr-spinel can be used as a powerful tracer for Cr-spinel metasomatic modification.

Our findings, therefore, provide a framework for a re-evaluation of Cr-spinel compositions in mantle rocks. They will also have numerous new applications. These include using Cr-spinel composition as a melt-rock interaction indicator, and as a tracer of mantle heterogeneity. In addition, Cr-spinel as a FME-carrying mineral may also provide a feeding mechanism for FME into the deep mantle chemical cycle because it breaks down at >1000 km depth[69]. The present findings will thus help to develop a new way of deciphering deep mantle metasomatism and heterogeneity through analysing non-traditional isotopes (e.g., Li, Zn, Ti, and Ni) in Cr-spinel.

Finally, we suggest that future studies of Cr-spinel chemistry need to first carry out systematic microanalyses in order to detect any heterogeneity of trace elements (including FME) using methods such as X-ray elemental mapping, EMP (at least 3–5 spots within the same grain spreading from the inner to outer core, and at least three grains per sample), and LA-ICPMS. More advanced methods such as EBSD and APT, plus careful interpretation of the obtained results, will help to detect modified/heterogeneous Cr-spinels. Whereas altered Cr-spinels may help to address post-melting metasomatic process, non-modified/homogenous ones can then be used for tracking partial melting.

## Methods

**Bulk rock chemical analysis**. Whole rock geochemistry of major and trace (including rare earth) elements were carried out for some samples of serpentinized peridotites (Supplementary Data 2; all whole-rock results are reported on a volatile-free basis). Whole-rock samples were crushed with a polyethylene-wrapped hammer into <0.5 cm pieces and then grounded with ethyl alcohol in an agate mill to grain sizes below 50 μm. Major element compositions were analyzed using a X-ray fluorescence spectrometer (Shimadzu, XRF-1800) at the Pukyong National University, South Korea. Analytical conditions were 40 kV accelerating voltage and 70 mA beam current. Analytical precision is better than 2% for major elements. All glass beads were analyzed three times and the averages are used. The trace and rare-earth elements of the studied samples were analyzed using an ICP-MS (VG Elemental Ltd., PQ3) at the Korean Basic Science Institute (KBSI) at Ohchang, South Korea. Acid digestion with hydrofluoric acid is routinely used to digest geological materials for the trace-element determination. A 100 mg of powdered sample was accurately weighed into the PTFE digestion vessel, added with 5 ml mixed acid (HF:HNO$_3$:HClO$_4$ = 4:4:1). The sample vessel was tightly capped and placed on a hot plate for 24 h at 190 °C and then cooled to room temperature. After opening the cap, the sample was subsequently evaporated to incipient dryness. The above process was repeated once more. The residue was dissolved in 10 ml of 1% HNO$_3$ with gentle heating, until a clear solution resulted. Determinations for USGS reference samples (BIR-1 and MUH-1) agree with recommended values within suggested tolerances. The precision of the measurements by repeated analyses of reference samples is better than ± 5% for trace elements.

**Electron probe micro analyses (EMPA)**. Quantitative chemical analyses of major and some minor elements in Cr-spinel were carried out using a JEOL JXA-8800 electron-probe at Kanazawa University, Japan (Supplementary Data 2). The analytical conditions were 20 kV accelerating voltage, 20 nA probe current and 3 μm beam diameter. A ZAF-correction was made to correct the raw data. Ferric and ferrous iron redistribution from EMP analyses was made using the charge balance equation of Droop[70]. Natural mineral standards were used for calibration: quartz for Si, corundum for Al, eskolaite for Cr, fayalite for Fe, periclase for Mg, manganosite for Mn, wollastonite for Ca, jadeite for Na and pentlandite for Ni. Si (<0.03 wt%) and Na (<0.02 wt%) for the studied Cr-spinel are below the detection limits.

**X-ray mapping of Cr-Spinel**. Al-Cr-Mg-Fe X-ray element distribution maps of Cr-spinel grains in the studied peridotites were acquired using a wavelength dispersive X-ray spectrometry (WDS) on a JEOL JXA-8800 electron-probe at Kanazawa University, Japan. The analytical conditions were 20 kV accelerating voltage, 20 nA probe current and <1 μm beam diameter. Dwell times on each spot were between 20 and 30 ms according to the analyzed element and the sensitivity of the detector.

**Electron back-scattered diffraction (EBSD)**. EBSD data were collected at the Microscopy and Microanalysis Facility, John de Laeter Centre, Curtin University, using a Tescan MIRA3 SEM with Oxford Instruments Symmetry EBSD detector. Data were collected at 20 kV and ~1 nA beam current, with an analytical step size of 2 μm. EBSD data were collected using Oxford Aztec 4.1 acquisition software. Data were noise reduced using a wildspike and 5 nearest neighbor zero solution

algorithm in Oxford Instruments Channel 5.12 software. Channel 5.12 was also used to create misorientation maps used to investigate the microstructural relationship between spinel core and rim. Relatively poor-indexing of the spinel represents the difficulty in polishing the analyzed grains after laser-ablation analysis.

**Laser ablation-inductively coupled plasma-mass spectrometry (LA-ICP-MS) micro analysis**. Trace element compositions of Cr-spinel (Supplementary Data 2) were determined using a 193 nm ArF Excimer LA-ICP-MS at Korean Basic Science Institute (KBSI), South Korea (Teledyne Cetac Technologies equipped with Analyte Excite). Analyses were performed by ablating 30–50 μm diameter spots at 10 Hz with an energy density of 5 J/cm$^2$ per pulse. Signal integration times were 60 s for a gas background interval and 60 for an ablation interval. The NIST SRM 612 glass was used as the primary calibration standard and was analyzed at the beginning of each batch of <5 unknowns, with a linear drift correction applied between each calibration. The element concentration of NIST SRM 612 for the calibration is selected from the preferred values of Pearce et al.[71]. Each analysis was normalized using $^{57}Fe$ as internal standard elements, based on Fe contents obtained by Electron probe micro analysis. All minerals were analyzed multiple times and the averages were used. The relative standard deviations (RSD) of the trace elements in the minerals were mostly 5–10%. We used the Glitter software to process the data, which shows ablation profiles for elements and permits to reduce analytical contamination. The LA-ICP-MS ablation profiles of the intensity (counting) vs time for all analyses of the studied Cr-spinel are provided in supplementary information (Supplementary Fig. 12). None of the calculated peaks has any significant spike signals for $^{29}Si$ and $^{57}Fe$, which indicates that no silicate inclusion and/or magnetite was measured. The slight difference between EPMA and LA-ICPMS data for trace elements such as Ni, Mn, and Ti is due to matrix effect and the different standards used, but the LA-ICPMS data is more precise than the EMPA data especially for trace elements.

**Atom probe tomography (APT)**. APT has the ability to characterize and visualize the 3D distribution of atoms at sub-nanometer resolution[72]. Recent advancements and the development of laser pulsing capabilities has made possible the analysis of non-conductive materials such as most rock-forming minerals. The technique is based on the field-evaporation of atoms from a needle-shaped specimen under a high electric field. The 3D position of atoms is given by the impact location on a position sensitive detector and the succession of detection events. The ions are identified using time-of-flight mass spectrometry by measuring the time between the laser pulse and the detection event. For more details about this technique see Ref. [73,74]. Four needle-shaped specimens (7 μm length and 2.5 μm depth for each one) for APT measurements were prepared with a Ga$^+$ Tescan Lyra3 focused ion beam coupled with a scanning electron microscope (FIB-SEM) at Curtin University following the lift out method[75]. The specimens were extracted from a polished thin section and their location is indicated on Fig. 1b (Supplementary Data 2). The FIB was operated at 30 kV during the manufacturing of the specimen and a low kV (2 kV) clean-up was implemented to remove the surface layer affected by high-energy Ga$^+$ ion implantation. APT measurements were performed on a CAMECA LEAP 4000X HR at Curtin University. The instrument was operated in laser mode ($\lambda = 355$ nm) with a laser pulse energy of 150 pJ and a pulse frequency of 200 kHz. The specimens were kept at 60 K during analysis with a detection rate maintained at 1 detection event every 100 laser pulses. The specimens yielded between 100 million atoms (M5, M6 and M10) and 109 million atoms (M8) (See Supplementary Movies 1 and 2 for Al in M6 and M10 specimens). The mass spectra were characterized by large peak tails similar to other oxides such as rutile[76]. In the mass spectra (Supplementary Fig. 13), the cations are present as different molecular species with singly-charged to triply-charged ions. For example, Cr is present as Cr$^+$, Cr$^{++}$, CrO$^+$, CrO$^{++}$, CrO$_2^+$, CrO$_2^{++}$, CrO$_3^+$ and Cr$_2$O$_3^+$; and Al is present as Al$^+$, Al$^{++}$, Al$^{+++}$, AlO$^+$, AlO$^{++}$, AlO$_2^+$, Al$_2$O$^{++}$, and AlCrO$_3^{++}$. The combination of multiple ionic species for each element and large thermal peak tails leads to difficulties in quantifying the composition of the analyzed nano-scale domains. The estimation of the O composition from atom probe data is also renowned to be difficult[77]. These limitations indicate that the composition calculated from the atom probe data will differ from expected stoichiometry and other techniques that apply standardization protocols (i.e., EPMA and LA-ICPMS). However, the method used for the calculation of the composition in this study was consistent across all datasets, indicating that the differences between atom probe specimens are reliable.

**Filter criteria for Cr-spinel data**. Based on the available data, we filtered out modified Cr-spinel results that (1) show Al-Cr reverse/ asymmetric zoning "heterogeneity" in the same grain, (2) have high Al and Cr content range >1.5 wt%, standard deviation >0.5 or >5% variability percentage within a single sample, and (3) have Cr# variability greater than 1% or standard deviation >0.01 within a single sample. In addition, we excluded samples that have Cr-spinel with only one analysis and no standard deviation or variability percentage.

## Data availability
All the data that are necessary for evaluating the findings of this study are available within this article and it's Supplementary Information.

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

## Acknowledgements

We thank A. Cipriani, D. Brunelli, M. Seyler, M. D'Errico, S. Mallick, L. Franz, J. Warren, D. Ionov, A. Bénard, Y. Ohara, and Y. Bai for sharing their research results and for

their helpful discussion, A. Abu El-Ela and A. Hassan are thanked for assistance with field work and regional geology, T. Morishita for assisting with the Electron Microprobe analyses, X-ray mapping and suggestions, Nguyen The Cong for helping with the LA-ICPMS analyses. Field work was supported by Tanta University, Egypt. Financial support by the Australian Research Council (grant FL150100133 to Z.X.L.) is acknowledged. This is a contribution to IGCP648: Supercontinent Cycles and Global Geodynamics.

## Author contributions

H.G. establish the idea, collect all the data, prepare the figures, and wrote the first draft of the manuscript. Y.K. did the LA-ICPMS analyses. D.F. and D.S. perform APT. S.R. did EBSD. M.H. help in collecting samples. S.A., L.S.D. and Z.X.L. designed the paper and clarify the concepts. All the authors participated in the interpretation of the results and preparation of the final version of the manuscript.

## Competing interests

The authors declare no competing interests.
