## [Peer Review File · Nature Communications]

Reviewers' comments:

Reviewer #1 (Remarks to the Author):

Is Cr-spinel a reliable petrogenetic indicator for mantle melts?

By Hamed Gamal EL Dien, Shoji Arai, Luc-Serge Doucet, Zheng-Xiang Li, Yongwoo Kil, Mohamed Hamdy

This paper presents Cr-spinel compositional data to demonstrate that 1) spinel composition can be altered by metasomatism and 2) spinel can host fluid-mobile trace elements. The authors use these observations to make two major points: 1) that Cr# heterogeneity in spinel (on a grain to sample scale) make them not useful as a petrogenetic indicator, but rather useful as a metasomatic indicator; and 2) that spinel that host FME indicate a subduction zone environment for the metasomatic overprinting.

They use elemental measurements on spinel from Neoproterozoic serpentinites from the Arabian-Nubian Shield to demonstrate the relationship between zonation of Cr-rich cores and Al-rich rims and FME trace elements. They then refer to a global compilation of fore-arc and mid-ocean ridge (MOR) spinel compositional data, which they re-evaluate for heterogeneities in spinel composition.

They conclude that spinel Cr# compositional overlap between fore-arc and mid-ocean ridge settings is too large to make it a useful tectonic indicator. They also state that they re-interpret previously reported spinel MOR data as the result of melt-rock interaction rather than partial melting.

In general, I think this is an excellent and intriguing contribution that has broad implications for the interpretation of spinel data in past and future studies and is appropriate for publication in Nature Communications. However, I think the it would benefit from reframing and/or clarification of the argument and modification to the figures before it is ready for acceptance. I have outlined my review below into Broad comments, Line comments and Figure comments.

Broad comments:

1. The main argument of this paper is that spinel Cr# (as it has been traditionally used) is not a good petrogenetic indicator because the assumptions that spinel records primary melting and does not experience significant post-formation alteration are invalid. The authors demonstrate the importance of documenting the chemical heterogeneity in spinel to properly interpret the melting vs metasomatic history, and they show that the classic interpretation of spinel as an indicator of primary mantle melting may not be valid in many cases.

While I am convinced from their data that spinel can be modified and may not be a reliable indicator for primary melting, I question whether the modified spinel cannot still be useful as a tectonic indicator, particularly for fore-arc settings. Couldn't depleted Cr# and modification (reverse zoning) be a diagnostic characteristic for fore-arc/subduction zone environments? For example, the tectonic setting described for the ANS peridotite is a fore-arc mantle wedge environment (Line 107), and when the ANS spinel Cr#'s are evaluated in the discussion by the 'conventional' approach it gives a fore-arc environment for their formation (Line 207).

Is the spinel chemistry really not a useful indicator of tectonic setting, or may it still be useful and what needs to change is how we interpret their chemistry?

2. In Line 69, the authors say that they are describing 'a new type of Cr-spinel zonation'. However, they already mentioned that reverse zoning can occur (Line 62) as a result of stress/deformation, element exchange or melt/rock interaction. Therefore, it seems misleading to label the reverse zoning as 'a new type of zonation'. What sounds new is that they interpret this zoning to be unrelated to the 'magmatic or metamorphic history' (Line 74). Instead they interpret the reverse zoning as an indicator of metasomatism in their samples, and go on to say that this reverse zoning is a widespread phenomenon at various tectonic settings.

This section reads a bit contradictory with the description of 'new zonation' and it being widespread / recognized previously, but interpreted differently. The message in this section should be re-written to more clearly present what is 'new' about their samples.

Related to this, it seems that the argument that the reverse zonation in the ANS spinel is the result of metasomatism and not deformation or metamorphism is critical. Although it is stated that the peridotites are 'non-deformed' there is not a thorough enough discussion of this in the geologic background presented in Lines 82-92. Same with the metamorphic history – considering the age of these, there should be a short summary statement of the tectonic/metamorphic history since emplacement. It does not have to be long, but enough to convince the reader that indeed deformation and metamorphism can be ruled out as the drivers for spinel modification.

3. As the text is currently written, the authors argue that spinel can be modified at all settings and therefore is not a reliable indicator of degree of melting or tectonic setting. I think they make a compelling argument for why spinel from fore-arc settings do not record primary melting signatures (although, their modification does seem like it could be a good indicator of tectonic setting). However, how this relates to MOR settings is not as clear. Although there is some discussion in the text (lines 240-256), there is not enough information presented to convincingly demonstrate that spinel Cr# isn't effective at recording the degree of melting or tectonic setting for MOR samples. If the authors want to make this point, then I suggest the data presented in the supplement should be worked into this part of the discussion and Figure 4. If MOR peridotite data could be filtered and not filtered (as was done with the fore-arc peridotites), this would be much more convincing.

Line comments:

Line 71: I would remove the second of "gold' standard (which is written as 'golden standard').

Line 82-93: This paragraph gives a brief background of the samples. As mentioned above, I think it requires some discussion of the metamorphic and deformation history, in addition to the mineralogy and melt history described.

Line 105: I am skeptical that LOI is a good indicator of whether the FME enrichment is from serpentinization. There are many cases where LOI and FME do not correlate, but the FME enrichment is still related to serpentinization (Deschamps et al 2013 review).

Line 117: heterogeneous spelled wrong

Line 170 (and Figure 3a-d): Is there any evidence that the Li, Rb, Sr, Cs enrichments are not related to inclusions of serpentine or some other microphase? Do these elements correlate with Si? Is there any evidence for inclusions (melt or mineral or fluid) in the spinel rims? Could this evidence be added to the supplement?

Line 172: Is Al added or Cr depleted (or both)? Can you tell?

Line 207: This paragraph could use some clarification. It begins by challenging that spinel preserves the partial melting history of mantle peridotites, in general. However, it only provides an example that is relevant to fore-arc/subduction zone settings (i.e., modification by slab fluids). It does not demonstrate how this would be a problem for MOR. Very good points are made about how core vs rim Cr# would give misleading melt % and plot in the fore-arc and MOR fields. However, the statement that the 'conventional' approach would give a misleading conclusion by plotting their samples in the fore-arc setting seems contradictory, since previously it is stated that the samples are from a fore-arc.

Line 232: This sentence is confusing... 'that contributed to metasomatism' – what is this phrase referencing?

Line 236-239: If the range spinel Cr#'s are plotted for modified and non-modified spinel, couldn't this still be a good tectonic indicator for fore-arcs? Or would a new set of criteria be required: i.e., Cr#, FME, zonation?

Line 265-268: This is one of the most compelling points about Cr# being an ineffective geotectonic indicator (that 50-75% of all data overlap in the same field). You should consider leading with this or adding it to the abstract. It looks as though some data plot out of the field? How were the 'lines' drawn?

Figure comments:

Fig 2: Plots f and g are confusing/difficult to read.

In 2f the colors of the data points make the plot too difficult to interpret. I assume C, B and R refer to Core, B??, and Rim? Instead of everything having a different color, I suggest making the the samples or cores vs rims color coordinated. For example, each sample could be a different color and C, B and

R a different shade. OR all the C's are one color, B's a different color, etc. and the samples are different shades. Whichever way pulls out the trend you are trying to emphasize better.

Plot 2g – Confusing to read. The legend doesn't appear to match the plotted data? Is that offset real?

Fig 3: Plot 3e can be modified to be easier to read.

As it is, the lines all overlap and make it difficult to see the variations. I suggest making the blue depleted spinel lines a lighter color or a little transparent or dashed. Since it has equal weight and is on top, it comes across as the most important data on the graph, when I think your data should be highlighted more strongly. Also, the main take-away from the text is that the spinel from this study more closely match subduction inputs than the depleted spinel from Szilas for FME's (Cs, Li, Rb, Sr). Perhaps highlight these elements or annotate the graph to point this out.

Fig 4: Why is each fore-arc setting separated out with its own color? Necessary? Also, Fig 4a isn't necessary. It's not cited in the text.

Reviewer #2 (Remarks to the Author):

Review of the manuscript : "Is Cr-spinel a reliable petrogenetic indicator for mantle melts ? " By Hamed Gamal El Dien et al.

General comments

In this manuscript, the authors present an impressive data set on Cr-spinel for a large spectrum of chemical elements including trace elements not determined in routine analyses. The covariation between Al content and fluid mobile elements in Cr-spinel is a robust, original and important result supporting the author's main conclusion. The general idea (i.e. Cr-spinel composition reflects secondary petrological processes, i.e. "metasomatism", rather than partial melting and/or fractional crystallization) is not really new (see below). However, the authors successfully demonstrate that the interaction with a fluid phase (or with a hydrated melt) is partly or fully responsible of the variations in spinel composition in residual mantle rocks. This result is important enough to deserve publication in a major journal like Nature. The section devoted to a re-evaluation of previous data on Cr-spinel is extremely interesting and convincing and contribute to the broad interest of the manuscript.

Detailed comments

L. 34-35. "The heterogeneity of Cr# in Cr-spinel can therefore be used as an excellent tracer for metasomatic modification, but not as a petrogenetic indicator." This sentence is somewhat confusing as metasomatic modification is a petrogenetic process per se. Accordingly the composition of Cr-spinel is an excellent petrogenetic indicator. Further in the text, it becomes clear that the authors restrict the concept of "petrogenetic indicator" to a proxy for the degree of partial melting, ... should they?

L. 44. Write "mantle melting conditions" rather than "mantle melt conditions".

L. 56. "Almost all authors (...)" and L. 278-280 "We also demonstrate that the wide range of Cr# of Cr-spinel in mantle rocks is a function of melt-rock interaction rather than partial melting". These sentences do not pay tribute to previous papers where it has been shown that the Cr# of Cr-spinel can be affected by metasomatism. The same zoning pattern (for major and minor elements only, not for trace elements I hast to say) as the one reported here was observed in Lanzo peridotite and attributed to the effect of melt-rock reactions triggered by shearing in high temperature mantle mylonites (Fig. 13a in Kaczmarek and Müntener, 2008). The effect of melt rock interaction on the composition of Cr-spinel was also documented in the Trinity peridotite (Fig. 5c in Ceuleneer and Le Sueur (2008) where we have shown that the compositional range of Cr-spinel is much broader along the walls of former melt channels than in peridotite collected away from melt channels). Ceuleneer (2004) also suggested that the range of degrees of partial melting of Oman peridotite estimated by Le Mee et al (2004) (a reference cited by the authors), on the basis, among others, of variations in the Cr# of Cr-spinel in mantle harzburgite of Oman, might be an effect of mantle-melt reaction (the "reactive filter hypothesis"). Abily and Ceuleneer (2013) and Rospabé et al. (2018) also present variations in Cr spinel composition in the shallow mantle section of the Oman ophiolite that they interpret in terms of modification by percolating melts/fluid. Many other references could be cited (sorry to refer mostly to publications from my group!).

L. 60-61. "(... characterized by Mg-Al-rich core (low Cr#) and Cr-Fe+2-rich rim (high Cr#)". This is correct but the way this sentence is written is potentially confusing for readers not familiar with the literature about Cr-spinel. These readers might think that Mg-Al-rich is synonymous of low Cr# and Cr-Fe-rich of high Cr# while the Cr# parameter is the Cr/(Al+Cr) atomic ratio). I know the authors know but the formulation might be misleading. I understand that the authors mean that a relationship generally exists between Mg, Fe, Cr and Al contents of Cr-spinel but this is far from being self-evident for a "generalist" reader.

L. 69. The authors should be more explicit and explain why they can claim that the type of zoning they determined is "new". It looks quite similar to the zoning determined by Bai et al (2018) (cited by the authors) in the North China craton, even if the interpretation of Bai et al differ (and I would tend to support the interpretation of the authors of the present ms.). It is also similar to the zoning

determined (along profiles, not by chemical mapping) in the publication of Kaczmarek and Müntener (2008).

L. 135. "The cores of Cr-spinels show a wide variation in compositions between grains from the same sample". This heterogeneity might be an artefact of the 3-D nature of the Cr-spinel grains. As a matter of fact, the thin sections randomly expose 2-D sections of the grains not necessarily reaching the very centre of these grains. In other words, the variations in the apparent cores (in 2-D) composition might just reflect zoning (in 3-D). The authors partially address this potential problem in the ms. (L. 203-206) but the arguments they present to exclude 3-D effect are quite light.

L. 182. "We interpret the Al, Cr and FME zoning in Cr-spinel to be the result of cryptic metasomatism by interactions of hydrous, Al-rich and silicic slab-derived melts with the host peridotites." The authors should precise which mechanism they have in mind (diffusion, recrystallization after corrosion, etc...?). This question is evoked in the manuscript but not developed enough, although it should be a key aspect of their demonstration.

L. 211. "Artificial" is not appropriate... metasomatism is a natural process, isn't?!

References cited in this review and not included in the reference list of the manuscript.

Abily B. and Ceuleneer G. The dunitic mantle-crust transition zone in the Oman ophiolite: Residue of melt-rock interaction, cumulates from high-MgO melts, or both? *Geology*, 41, 67-70, 2013.

Ceuleneer G. Mantle mapped in the desert. *Nature*, 432, 156-157, 2004.

Ceuleneer G. and Le Sueur E. The Trinity ophiolite (California): the strange association of fertile mantle peridotite with ultra-depleted crustal cumulates. *Bull. Soc. Geol. France*, 179, 5, 503-518, 2008.

Kaczmarek M.-A. and Müntener O. Juxtaposition of melt impregnation and high temperature shear zones in the upper mantle; field and petrological constraints from the Lanzo peridotite (Northern Italy). *Journal of Petrology*, 49, 2187-2220, 2008.

Rospabé M., Benoit M., Ceuleneer G., Hodel F. and Kaczmarek M.-A. Extreme geochemical variability through the dunitic transition zone of the Oman ophiolite: implications for melt/fluid-rock reactions at Moho level beneath oceanic spreading centers. *Geochim. Cosmochim. Acta*, 234, 1-23, 2018.

Georges Ceuleneer, Toulouse, January 29th 2019

Reviewer #3 (Remarks to the Author):

Background: This article challenges a well established notion in the field of mantle petrology, that Cr-spinel may not be the ideal carrier of information relevant to the extent of mantle melting. Cr-spinel is chemically zoned in many locations worldwide, and its composition may reflect metasomatic processes rather than degree of melting.

Recommendation: The ideas developed in this work are interesting and certainly worthwhile exploring (i.e. are many of the Cr-spinels from ultramafic rocks zoned?). The data does a great job at supporting the idea that THEIR samples have indeed undergone complex processes after mantle melting, and that Cr-spinel compositions may reflect those secondary rather than primary processes. The key question here is: is this true of most ultramafic rocks inferred to provide records of melting extent? I think that unfortunately the arguments proposed by the authors are largely insufficient to substantiate this claim, and am therefore unsure the part that is conclusive (that their Spinel chemistry does record metasomatism rather than melting degree) warrants the type of significance usually accorded by Nature journals. If the authors can better corroborate the validity of these results to peridotites worldwide, this work should be considered for publication after resubmission. But not in its current state.

Main comments:

(1) Is Cr-spinel worldwide zoned, and is this zoning related to mantle melting or metasomatism?

The problem of the robustness of Cr-spinel as a melting recorder exposed in this contribution is contingent on demonstrating the ubiquity of heavily zoned, 'problematic' crystals worldwide. Using a compilation of spinel encountered in the literature, the authors claim to have substantiated widespread zoning. This compilation, and the associated statements in the text are, however, misleading. When I went to check Table S1, it became rapidly clear that the purported occurrences have in many cases nothing to do with spinel that would be used by mantle petrologists to examine extent of melting processes. For instance, (1) 28/63 are reported to be 'unclear in the paper' as to whether zoning occurs or not, I wonder what those bring to the table. (2) 8/63 examples are from basalt; those likely crystallized spinel after their extraction from their melting source, and have other mixing and cooling histories that create spinel zoning. No one would use those to get degree of melting in the manner of Hellebrand or others! (3) Chromitites (9/63), same issue, the examination of this type of spinel formation for primary signal on degree of melting wouldn't be exactly common! (4) At least 4/63 (didn't check all the references, e.g. Barth, Ozawa, Akmaz, Bai) have spinel that may be chemically variable as a result of deformation and again, not the candidates one would use for

the purposes stated in the paper. I didn't check the remaining references (10/63) but the point is, this compilation is misleading yet a vital component of the argument this paper is attempting to make.

(2) Sectioning effects and spinel cores

I think the author's point about zoning overprinting some of the initial Cr/Al composition is a valid one, and something that petrologists should be cautious with during analysis. However, if one analyzes the cores of the spinel shown in Fig. 1, the Cr does not vary too significantly. Another issue to be wary of is that for complex morphology zoned crystals like that shown in Fig. 1d is sectioning effects: it's possible that crystal was simply sectioned closer to its rim in the 3rd dimension and thus does not show its true core composition. Again, it doesn't invalidate completely the author's point about zoning, but it calls for more caution when seeing crystals that are obviously too resorbed or complexly shaped to trust for analytical work. It's critical to ask oneself if this is the sort of spinel one would use to extract mantle melting information.

Finally, I did not spend the time to make detailed edits but the ms and supplementary material is full of typos, the authors should make a thorougher read through before resubmitting.

Point-by-point response to reviewer's comments

(Original reviewer's comments are in black, and our responses are in blue. Line numbers refer to line numbers in the revised clean version.)

Reviewer #1 (Remarks to the Author):

Is Cr-spinel a reliable petrogenetic indicator for mantle melts?

By Hamed Gamal EL Dien, Shoji Arai, Luc-Serge Doucet, Zheng-Xiang Li, Yongwoo Kil, Mohamed Hamdy

This paper presents Cr-spinel compositional data to demonstrate that 1) spinel composition can be altered by metasomatism and 2) spinel can host fluid-mobile trace elements. The authors use these observations to make two major points: 1) that Cr# heterogeneity in spinel (on a grain to sample scale) make them not useful as a petrogenetic indicator, but rather useful as a metasomatic indicator; and 2) that spinel that host FME indicate a subduction zone environment for the metasomatic overprinting.

They use elemental measurements on spinel from Neoproterozoic serpentinites from the Arabian-Nubian Shield to demonstrate the relationship between zonation of Cr-rich cores and Al-rich rims and FME trace elements. They then refer to a global compilation of fore-arc and mid-ocean ridge (MOR) spinel compositional data, which they re-evaluate for heterogeneities in spinel composition.

They conclude that spinel Cr# compositional overlap between fore-arc and mid-ocean ridge settings is too large to make it a useful tectonic indicator. They also state that they re-interpret previously reported spinel MOR data as the result of melt-rock interaction rather than partial melting.

In general, I think this is an excellent and intriguing contribution that has broad implications for the interpretation of spinel data in past and future studies and is appropriate for publication in Nature Communications. However, I think it would benefit from reframing and/or clarification of the argument and modification to the figures before it is ready for acceptance. I have outlined my review below into Broad comments, Line comments and Figure comments.

Broad comments:

1. The main argument of this paper is that spinel Cr# (as it has been traditionally used) is not a good petrogenetic indicator because the assumptions that spinel records primary melting and does not experience significant post-formation alteration are invalid. The authors demonstrate the importance of documenting the chemical heterogeneity in spinel to properly interpret the melting vs metasomatic history, and they show that the classic interpretation of spinel as an indicator of primary mantle melting may not be valid in many cases.

While I am convinced from their data that spinel can be modified and may not be a reliable indicator for primary melting, I question whether the modified spinel cannot still be useful as a tectonic indicator, particularly for fore-arc settings. Couldn't depleted Cr# and modification (reverse zoning) be a diagnostic characteristic for fore-arc/subduction zone environments? For example, the tectonic setting described for the ANS peridotite is a fore-arc mantle wedge environment (Line 107), and when the ANS spinel Cr#'s are evaluated in the discussion by the 'conventional' approach it gives a fore-arc environment for their formation (Line 207).

Is the spinel chemistry really not a useful indicator of tectonic setting, or may it still be useful and what needs to change is how we interpret their chemistry?

Reply: Thanks for the comment which helped us to further clarify our argument. Our data and data compiled from the literature show that chemical reverse zoning is a ubiquitous feature that can be found in both arc-related peridotites and MOR peridotites. As such, we argue that the Cr-spinel chemistry cannot be used as a tectonic indicator but can be very useful for tracing metasomatism processes. This point has been discussed in details in the revised text (see lines 327-340). Moreover, we argue that reverse zoning are not diagnostic features for arc-related peridotites; because the same feature are also found in MOR-peridotites (please see lines 302-308 and Table S1).

2. In Line 69, the authors say that they are describing ‘a new type of Cr-spinel zonation’. However, they already mentioned that reverse zoning can occur (Line 62) as a result of stress/deformation, element exchange or melt/rock interaction. Therefore, it seems misleading to label the reverse zoning as ‘a new type of zonation’. What sounds new is that they interpret this zoning to be unrelated to the ‘magmatic or metamorphic history’ (Line 74). Instead they interpret the reverse zoning as an indicator of metasomatism in their samples, and go on to say that this reverse zoning is a widespread phenomenon at various tectonic settings.

This section reads a bit contradictory with the description of ‘new zonation’ and it being widespread / recognized previously, but interpreted differently. The message in this section should be re-written to more clearly present what is ‘new’ about their samples.

Reply: Thanks for pointing this out. This point has been clarified (see lines 72-84).

Related to this, it seems that the argument that the reverse zonation in the ANS spinel is the result of metasomatism and not deformation or metamorphism is critical. Although it is stated that the peridotites are ‘non-deformed’ there is not a thorough enough discussion of this in the geologic background presented in Lines 82-92. Same with the metamorphic history – considering the age of these, there should be a short summary statement of the tectonic/metamorphic history since emplacement. It does not have to be long, but enough to convince the reader that indeed deformation and metamorphism can be ruled out as the drivers for spinel modification.

Reply: Thanks for the comment. Deformation and metamorphic effects on Cr-spinel modification has now been discussed in details (see lines 159-178). We have added new EBSD images to decipher the effect of microstructures and stress on Cr-spinel. We also used covariation between Al and FME vs. $Fe^{3+}/(Fe^{3+} + Fe^{2+})$, which is widely used as an indicator for Cr-spinel metamorphic modifications (Barnes, 2000). The new results show clearly that the observed Cr-spinel modification is not related to either deformation or metamorphism.

3. As the text is currently written, the authors argue that spinel can be modified at all settings and therefore is not a reliable indicator of degree of melting or tectonic setting. I think they make a compelling argument for why spinel from fore-arc settings do not record primary melting signatures (although, their modification does seem like it could be a good indicator of tectonic setting). However, how this relates to MOR settings is not as clear. Although there is some discussion in the text (lines 240-256), there is not enough information presented to convincingly demonstrate that spinel Cr# isn’t effective at recording the degree of melting or tectonic setting for MOR samples. If the authors want to make this point, then I suggest the data presented in the supplement should be worked into this part of the discussion and Figure 4. If MOR peridotite data could be filtered and not filtered (as was done with the fore-arc peridotites), this would be much more convincing.

Reply: Thanks for the comment that gives us an opportunity to discuss the Al-Cr heterogeneity of Cr-spinel in MOR-peridotites. In addition, we have added a newly compiled dataset for MOR-peridotites (See Tables S1 and S3, and Fig. S1). Some of the original data and

discussions have now been transferred from the Supplementary Information to the main text and are discussed in more details (see lines 250-326). Cr-spinel from MOR-peridotites has been filtered into modified and non-modified, and the data are presented in Fig. 6.

Line comments:

Line 105: I am skeptical that LOI is a good indicator of whether the FME enrichment is from serpentinization. There are many cases where LOI and FME do not correlate, but the FME enrichment is still related to serpentinization (Deschamps et al 2013 review).

Reply: This point has been clarified. See lines 107-115 and Fig. S7.

Line 170 (and Figure 3a-d): Is there any evidence that the Li, Rb, Sr, Cs enrichments are not related to inclusions of serpentine or some other microphase? Do these elements correlate with Si? Is there any evidence for inclusions (melt or mineral or fluid) in the spinel rims? Could this evidence be added to the supplement?

Reply: We have now applied Atom Probe Tomography (APT) advanced technique to our work, which has the ability to characterize and visualize the 3D distribution of individual atoms (see method for more details about this technique). The outcome reinforces our interpreted 3D Al-Cr heterogeneity. APT results show that the homogenous distribution of Al, Mg and Fe. The non-detection of Si (as well as in EMPA <0.01) and absence of any isolated clusters across the specimens indicate that the high FME concentration in Cr-spinel is inherited and unrelated to silicates inclusions, e.g., serpentine phases (see line 192-208 and Fig. 5). This is the first time the APT technique has been applied to such study.

Line 207: This paragraph could use some clarification. It begins by challenging that spinel preserves the partial melting history of mantle peridotites, in general. However, it only provides an example that is relevant to fore-arc/subduction zone settings (i.e., modification by slab fluids). It does not demonstrate how this would be a problem for MOR. Very good points are made about how core vs rim Cr# would give misleading melt % and plot in the fore-arc and MOR fields. However, the statement that the 'conventional' approach would give a misleading conclusion by plotting their samples in the fore-arc setting seems contradictory, since previously it is stated that the samples are from a fore-arc.

Reply: This point has been modified and clarified. See lines 209-226 and 260-340.

Line 236-239: If the range spinel Cr#'s are plotted for modified and non-modified spinel, couldn't this still be a good tectonic indicator for fore-arcs? Or would a new set of criteria be required: i.e., Cr#, FME, zonation?

Reply: Please see the reply for point # 1 above.

Figure comments:

Fig 2: Plots f and g are confusing/difficult to read.

In 2f the colors of the data points make the plot too difficult to interpret. I assume C, B and R refer to Core, B??, and Rim? Instead of everything having a different color, I suggest making the the samples or cores vs rims color coordinated. For example, each sample could be a different color and C, B and R a different shade. OR all the C's are one color, B's a different color, etc. and the samples are different shades. Whichever way pulls out the trend you are trying to emphasize better.

Plot 2g – Confusing to read. The legend doesn't appear to match the plotted data? Is that offset real?

Reply: The figure has been modified as recommended. For Fig. 2g, the data from mantle peridotite xenoliths are from Ionov et al. (2011) and Franz et al. (2002).

Fig 3: Plot 3e can be modified to be easier to read.

As it is, the lines all overlap and make it difficult to see the variations. I suggest making the blue depleted spinel lines a lighter color or a little transparent or dashed. Since it has equal weight and is on top, it comes across as the most important data on the graph, when I think your data should be highlighted more strongly. Also, the main take-away from the text is that the spinel from this study more closely match subduction inputs than the depleted spinel from Szilas for FME's (Cs, Li, Rb, Sr). Perhaps highlight these elements or annotate the graph to point this out.

Reply: Thanks for the comment. The figure has been modified as recommended

Fig 4: Why is each fore-arc setting separated out with its own color? Necessary? Also, Fig 4a isn't necessary. It's not cited in the text.

Reply: The figure has been modified as recommended

Reviewer #2 (Remarks to the Author):

Review of the manuscript : “Is Cr-spinel a reliable petrogenetic indicator for mantle melts ? “
By Hamed Gamal El Dien et al.

General comments

In this manuscript, the authors present an impressive data set on Cr-spinel for a large spectrum of chemical elements including trace elements not determined in routine analyses. The covariation between Al content and fluid mobile elements in Cr-spinel is a robust, original and important result supporting the author's main conclusion. The general idea (i.e. Cr-spinel composition reflects secondary petrological processes, i.e. “metasomatism”, rather than partial melting and/or fractional crystallization) is not really new (see below). However, the authors successfully demonstrate that the interaction with a fluid phase (or with a hydrated melt) is partly or fully responsible of the variations in spinel composition in residual mantle rocks. This result is important enough to deserve publication in a major journal like Nature. The section devoted to a re-evaluation of previous data on Cr-spinel is extremely interesting and convincing and contribute to the broad interest of the manuscript.

Detailed comments

L. 34-35. “The heterogeneity of Cr# in Cr-spinel can therefore be used as an excellent tracer for metasomatic modification, but not as a petrogenetic indicator.” This sentence is somewhat confusing as metasomatic modification is a petrogenetic process per se. Accordingly the composition of Cr-spinel is an excellent petrogenetic indicator. Further in the text, it becomes clear that the authors restrict the concept of “petrogenetic indicator” to a proxy for the degree of partial melting, ... should they?

Reply: Thanks for the comment. The sentence was modified. See lines 33-40.

L. 56. “Almost all authors (...)” and L. 278-280 “We also demonstrate that the wide range of Cr# of Cr-spinel in mantle rocks is a function of melt-rock interaction rather than partial melting”. These sentences do not pay tribute to previous papers where it has been shown that the Cr# of Cr-spinel can be affected by metasomatism. The same zoning pattern (for major and minor elements only, not for trace elements I hast to say) as the one reported here was observed

in Lanzo peridotite and attributed to the effect of melt-rock reactions triggered by shearing in high temperature mantle mylonites (Fig. 13a in Kaczmarek and Müntener, 2008). The effect of melt rock interaction on the composition of Cr-spinel was also documented in the Trinity peridotite (Fig. 5c in Ceuleneer and Le Sueur (2008) where we have shown that the compositional range of Cr-spinel is much broader along the walls of former melt channels than in peridotite collected away from melt channels). Ceuleneer (2004) also suggested that the range of degrees of partial melting of Oman peridotite estimated by Le Mee et al (2004) (a reference cited by the authors), on the basis, among others, of variations in the Cr# of Cr-spinel in mantle harzburgite of Oman, might be an effect of mantle-melt reaction (the “reactive filter hypothesis”). Abily and Ceuleneer (2013) and Rospabé et al. (2018) also present variations in Cr spinel composition in the shallow mantle section of the Oman ophiolite that they interpret in terms of modification by percolating melts/fluid. Many other references could be cited (sorry to refer mostly to publications from my group!).

Reply: Thanks for the comment. These relevant references have been added to the revised version of the manuscript.

L. 60-61. “(... characterized by Mg-Al-rich core (low Cr#) and Cr-Fe+2-rich rim (high Cr#)”. This is correct but the way this sentence is written is potentially confusing for readers not familiar with the literature about Cr-spinel. These readers might think that Mg-Al-rich is synonymous of low Cr# and Cr-Fe-rich of high Cr# while the Cr# parameter is the Cr/(Al+Cr) atomic ratio). I know the authors know but the formulation might be misleading. I understand that the authors mean that a relationship generally exists between Mg, Fe, Cr and Al contents of Cr-spinel but this is far from being self-evident for a “generalist” reader.

Reply: The sentence has been modified. See lines 65-68.

L. 69. The authors should be more explicit and explain why they can claim that the type of zoning they determined is “new”. It looks quite similar to the zoning determined by Bai et al (2018) (cited by the authors) in the North China craton, even if the interpretation of Bai et al differ (and I would tend to support the interpretation of the authors of the present ms.). It is also similar to the zoning determined (along profiles, not by chemical mapping) in the publication of Kaczmarek and Müntener (2008).

Reply: The sentence has been modified. See lines 72-84.

L. 135. “The cores of Cr-spinels show a wide variation in compositions between grains from the same sample”. This heterogeneity might be an artefact of the 3-D nature of the Cr-spinel grains. As a matter of fact, the thin sections randomly expose 2-D sections of the grains not necessarily reaching the very centre of these grains. In other words, the variations in the apparent cores (in 2-D) composition might just reflect zoning (in 3-D). The authors partially address this potential problem in the ms. (L. 203-206) but the arguments they present to exclude 3-D effect are quite light.

Reply: Thanks for the comment. In the revised version, we used the new Atom Probe method (see methods for more details) for the first time in mantle petrology studies to more robustly document the 3D Al-Cr heterogeneity between core and rim of Cr-spinel crystals. See lines 192-208 and Fig. 5.

L. 182. “We interpret the Al, Cr and FME zoning in Cr-spinel to be the result of cryptic metasomatism by interactions of hydrous, Al-rich and silicic slab-derived melts with the host peridotites.” The authors should precise which mechanism they have in mind (diffusion, recrystallization after corrosion, etc...?). This question is evoked in the manuscript but not developed enough, although it should be a key aspect of their demonstration.

Reply: This section has been modified. See lines 156-158.

L. 211. “Artificial” is not appropriate... metasomatism is a natural process, isn't?!

Reply: Thanks for the comment. The sentence has been modified.

Georges Ceuleneer, Toulouse, January 29th 2019

Thanks for these comments that helped us to clarify some points.

Reviewer #3 (Remarks to the Author):

Background: This article challenges a well established notion in the field of mantle petrology, that Cr-spinel may not be the ideal carrier of information relevant to the extent of mantle melting. Cr-spinel is chemically zoned in many locations worldwide, and its composition may reflect metasomatic processes rather than degree of melting.

Recommendation: The ideas developed in this work are interesting and certainly worthwhile exploring (i.e. are many of the Cr-spinels from ultramafic rocks zoned?). The data does a great job at supporting the idea that THEIR samples have indeed undergone complex processes after mantle melting, and that Cr-spinel compositions may reflect those secondary rather than primary processes. The key question here is: is this true of most ultramafic rocks inferred to provide records of melting extent? I think that unfortunately the arguments proposed by the authors are largely insufficient to substantiate this claim, and am therefore unsure the part that is conclusive (that their Spinel chemistry does record metasomatism rather than melting degree) warrants the type of significance usually accorded by Nature journals. If the authors can better corroborate the validity of these results to peridotites worldwide, this work should be considered for publication after resubmission. But not in its current state.

Main comments:

(1) Is Cr-spinel worldwide zoned, and is this zoning related to mantle melting or metasomatism?

The problem of the robustness of Cr-spinel as a melting recorder exposed in this contribution is contingent on demonstrating the ubiquity of heavily zoned, ‘problematic’ crystals worldwide. Using a compilation of spinel encountered in the literature, the authors claim to have substantiated widespread zoning. This compilation, and the associated statements in the text are, however, misleading. When I went to check Table S1, it became rapidly clear that the purported occurrences have in many cases nothing to do with spinel that would be used by mantle petrologists to examine extent of melting processes. For instance, (1) 28/63 are reported to be ‘unclear in the paper’ as to whether zoning occurs or not, I wonder what those bring to the table. (2) 8/63 examples are from basalt; those likely crystallized spinel after their extraction from their melting source, and have other mixing and cooling histories that create spinel zoning. No one would use those to get degree of melting in the manner of Hellebrand or others! (3) Chromitites (9/63), same issue, the examination of this type of spinel formation for primary signal on degree of melting wouldn't be exactly common! (4) At least 4/63 (didn't check all the references, e.g. Barth, Ozawa, Akmaz, Bai) have spinel that may be chemically variable as a result of deformation and again, not the candidates one would use for the purposes stated in the

paper. I didn't check the remaining references (10/63) but the point is, this compilation is misleading yet a vital component of the argument this paper is attempting to make.

Reply: Thanks for this comment that gave us the opportunity to more systematically compile datasets from present-day oceanic peridotites and some examples from on-land ophiolites and mantle xenoliths. These data further demonstrate that Cr-spinel from peridotites is widely modified and its Al-Cr heterogeneity reflects melt/rock interaction rather than primary melting processes (see lines 236-326 and Table S1). Table S1 is now modified to show only peridotite data.

(2) Sectioning effects and spinel cores

I think the author's point about zoning overprinting some of the initial Cr/Al composition is a valid one, and something that petrologists should be cautious with during analysis. However, if one analyzes the cores of the spinel shown in Fig. 1, the Cr does not vary too significantly. Another issue to be wary of is that for complex morphology zoned crystals like that shown in Fig. 1d is sectioning effects: it's possible that crystal was simply sectioned closer to its rim in the 3rd dimension and thus does not show its true core composition. Again, it doesn't invalidate completely the author's point about zoning, but it calls for more caution when seeing crystals that are obviously too resorbed or complexly shaped to trust for analytical work. It's critical to ask oneself if this is the sort of spinel one would use to extract mantle melting information.

Reply: Thanks for the comment. We agree in general that much care is need when using Cr-spinel to diagnose mantle melting processes. In the revised version, we more robustly documented the 3D effect of Al-Cr heterogeneity using a recent advanced technique, Atom probe spectroscopy, for the first time in all mantle petrology studies. The data confirm that Al-Cr heterogeneity between core and rim of the studied Cr-spinel crystals. Please see lines 192-208 and Fig. 5.

Finally, I did not spend the time to make detailed edits but the ms and supplementary material is full of typos, the authors should make a thorougher read through before resubmitting.

Reply: Thanks for the comment. The manuscript has been proofread again to avoid the typos.

Reviewers' comments:

Reviewer #1 (Remarks to the Author):

Review of the manuscript : “Is Cr-spinel a reliable petrogenetic indicator for mantle melts ? “ By Hamed Gamal El Dien et al.

I reviewed a previous version of this study and my impression was globally positive in spite of a few weaknesses. The present version has been significantly improved.

. New types of analyses (EBSD and Atom Probe Tomography) have been performed. These new data increase the strength of the authors' arguments in favour of their working hypothesis: Cr-spinel present in their samples from the Arabian-Nubian shield has clearly suffered partial re-equilibration with metasomatic melts/fluids.

. The lattice misorientation observed in a few grains allows the authors to discard solid-state plastic deformation as a - frequently invoked - modification mechanism of Cr-spinel composition.

. APT data contribute to further document the reverse zoning in Cr-spinel composition that was revealed through more classical analytical methods (EPMA, LA-ICP-MS,...).

. The microscopic mechanisms envisioned for the modification process (essentially the combined effects of corrosion and diffusion) are now explicitly addressed.

. The potential bias related to geometrical effects (i.e. 2-D sections to analyse compositional variations in a 3-D mineral) is now discussed.

. The section devoted to a re-evaluation of published data on Cr-spinel was also highly improved as the authors detail the way they have handled this huge amount of data. This exercise was far from being trivial given the heterogeneous nature of the published information, e.g. the sampling and analytical strategies and methods, the amount and quality of the meta-data associated to the chemical analyses, as the field and petrographic descriptions, the part (core vs. rim) of the crystal that was analysed, etc... This synthesis is impressive and definitely contributes to broaden the interest of the manuscript. The authors demonstrate that mantle peridotite are, in most occurrences worldwide, affected by melt/fluid/rock interaction (“metasomatism”) post-dating the decompression melting event.

. A better tribute is made to previous advocates of mantle metasomatism (although a reference to Kaczmarek and Müntener (2008)(*) is still lacking. These authors documented reverse zoning in Cr-Al in Cr-spinel and discussed its significance in the frame of melt percolation in peridotite.

. The minor problems have been addressed.

In conclusion, this is a great piece of work that will likely be highly cited by the students of mantle processes. My opinion is that this work is now ready for publication.

(*) Kaczmarek M.-A. and Müntener O. Juxtaposition of melt impregnation and high temperature shear zones in the upper mantle; field and petrological constraints from the Lanzo peridotite (Northern Italy). *Journal of Petrology*, 49, 2187-2220, 2008.

Georges Ceuleneer

Reviewer #2 (Remarks to the Author):

This revised paper presents compelling data to show that Cr-spinel grains are susceptible to cryptic metasomatism that makes their utility as a melt-indicator at worst ineffective and at best questionable. The authors present rigorous characterization of a set of samples from an ancient ophiolite from the Arabian-Nubian Shield to show the relationship between Al-rich rims and FME, showing the effects of fluid-rock interactions with metasomatic fluids, likely in a fore-arc (FA) environment. They also use published data from MOR, FA and Back Arc environments to show that spinel heterogeneity is present in all these settings, and that significant overlap exists between the settings in terms of their Cr# - making the Cr# relatively ineffective as a tectonic indicator.

I believe this paper makes a worthy contribution that will have a large impact on the field and generate a lot of discussion and review of past studies. The authors have done significant work to improve the manuscript from the first submission. I have three remaining issues that need to be addressed before publication:

1. There are several spots in the paper where the authors make statements that imply spinel Cr# are suspect – and it is not clear to me whether they mean this as a blanket statement or in a more nuanced way. If it is their intention to say that all Cr# interpretations are suspect (as it comes across to me as I read it, e.g., Line 36), then it's my opinion that this statement is too strong for the data they present. They have really good evidence for their locality and a good case for fore-arc or subduction related peridotite. They can suggest and hypothesize that the Cr# can be an issue for MOR peridotites based on compilation data (which they do), but there remains the possibility that in

some cases the spinel do not violate the assumptions in the mantle-melting models and can still be used as a good indicator. I think the main take-away from this work is that there is strong evidence for cryptic metasomatism that requires in-depth characterization to detect. And only once homogeneity is proven can the spinel be used for mantle melting indices. Many historical studies may have not done this, but it does not mean they are invalid.

It also is written in a way that suggests that almost no one has worried about spinel heterogeneity before – e.g., in the abstract it says that almost all mantle petrologists have used this as a tool assuming a homogeneous, unaltered spinel. However, at several points through the paper they reference studies that have identified and discussed spinel zonation. It's possible that no previous studies have taken spinel zonation seriously enough, or interpreted it correctly in the view of the these authors, and that should be clarified.

Overall, both of these comments refer to the tone of the paper, which comes across too strongly in some spots and can be readily addressed by rephrasing and relatively minor editing.

2. I like the revision of the conclusions and implications. I would like them to add a sentence or two that states what future studies will need do to properly characterize spinel grains based on their findings. (e.g., BSE-EDS mapping, microprobe, # of spots, etc).

3. Lastly, there are a few spots in the text that require some clarification. I've noted them in the Line Comments below:

Line 36: "making Cr # ineffective as a geotectonic and mantle melting indicator" I am not sure whether the authors mean to imply this is true for all Cr# measurements, or only for those in spinel that experienced metasomatism as evidenced by the reverse Cr-Al zonation. If the former, then I do not think they fully proved this and should rephrase the sentence in a way that is not so definitive. If the latter, then 'in these cases' should be added to this clause to clarify that the authors are referring to the reversed zoned spinel and not all spinel Cr#s.

Line 68: "such reverse zoning is undetectable under microscope" – The way this is phrased implies that the other types of zoning are detectable. Certainly, in some cases (e.g., ferrite chromite rims), but often it's not detectable. Clarify the sentence.

Line 75: "describe a new type of Cr-spinel reverse zoning" – It's still unclear to me that this reverse zoning is "new" compared to the reverse zoning in the previous paragraph. Is the 'new' part the

process that caused it? Because it sounds like it's the same type of reverse zoning – Al/Mg rich rim, Fe/Cr rich core? If it is the process that makes it new, then to avoid confusion, the 'new' should be added in the next sentence which describes the process.

Line 202: This is a big one - They state that Fe and Mg content of the cores and rims analyzed by atom probe are relatively constant when compared to the variation in Al and Cr. However, looking at Fig 5, all four elements vary by <1 to ~2 at%, so this presentation of the data and the statement are not sufficiently convincing. It seems that there is major zonation recorded in Fig 1 (by microprobe?). What is the explanation for the discrepancy in the magnitude of difference between the core and rim concentrations measured in wt% presented in Fig 1 and atom% presented in Fig 5?

Line 218: "would point to a transition of forearc/MOR setting", this phrasing is confusing.

Line 226: "should not use Cr-spinel data... in such peridotite" – What is meant by "such peridotite"? Any peridotite that has zoned spinel, or in particular ophiolitic peridotites that have an unclear origin?

Line 363: "Our approach will also allow..." – What 'approach' that will allow mantle petrologists to use non-traditional isotopes are you referring to? This paper does not so much present an approach as it presents data to demonstrate the heterogeneity common to spinel grains. Please clarify the sentence.

Reviewer #3 (Remarks to the Author):

The main point of this manuscript is to demonstrate that Cr-spinels in mantle peridotites can be modified by post-melting processes, leading to an Al-Cr reverse zoning. According to the authors, this is a widespread process, and the post-melting modification distorts the inferred degree of partial melting commonly obtained from only measuring the spinel cores.

I disagree with this inference, and think the manuscript is flawed on several levels. There are issues with the data quality obtained on the sample presented in this study. In addition, the filtering process the authors use to emphasize the global importance of spinel heterogeneity is biased, not

taking into account analytical errors of published data, not considering modern concepts of the origin of length scales of heterogeneity in the upper mantle, and a general overextrapolation when information is absent. For these reasons alone, I recommend waiting with the publication of these findings until all issues are resolved. The conclusions and implications are invalid.

The reverse zoning of Cr-spinel is actually an interesting observation, often seen in plagioclase peridotites, but it does not merit publication in this journal, even after extensive revisions and additional analyses. Perhaps the authors are on to something interesting, as there are indeed a couple of examples from the literature that have reverse spinel zoning. However, these are examples of non-residual peridotites, for which estimates of their degree of melting would be pointless to begin with, so the scope of the manuscript would have to change into a more positive message. In its current form, the underlying tone is too negative and passive aggressive, trying to persuade the reader that there's a global problem resulting from peridotite researchers not working carefully.

[A] Data quality

(1) Comparing the Cr-spinel data obtained by EPMA and LA-ICP-MS for Mn, Ti and Ni shows that there is a systematic disagreement between the datasets. The laser data always yield much higher concentrations than the more reasonable looking EPMA data.

[see plots in supplement]

The LA-ICPMS data on the spinel were calibrated using NIST SRM 612, which is a silicate glass. Since the Cr-spinels have no Si, iron isotope ^{57}Fe was used as an internal reference mass. The problem is, in my opinion, that the low abundance of Fe (0.02% +/- 0.01% Fe_2O_3 according to the recommended values in Pearce et al 1997, used here. Instead, please refer to Jochum et al 2011, who report 51 +/- 2 ppm Fe) in the NIST standard glass introduces an enormous uncertainty in the absolute trace element concentration of the measured spinels, and the data cannot be used at all. Did the measurements perhaps include Al? If so, Al may be used as an internal reference mass in the offline data reduction, and the data may still be salvaged. A troublesome byproduct of using ^{57}Fe as reference mass is the incorporation of magnetite veins by the big laser beam.

(2) Ignoring the trace element accuracy for a second, the fluid-mobile elements Li, Sr, Rb, Cs appear to be enriched in the Al-rich domains, which dominantly occur at the spinel rims (Fig. 3). At first glance, this looks like a novel and fundamentally important observation. Unfortunately, any careful peridotite analyst knows that these data are not likely a high-temperature signature, but instead artefacts of mixed analyses of minute domains altered at low-temperature, incorporated by the large laser beam. I'd love to be corrected by the authors, but with the results and images

presented here, I'm going to be conservative, requiring more rigorous evidence to be convinced that this is indeed a real high-temperature signal.

(a) Was Si acquired by LA-ICPMS? Were there any Si- or Fe-spikes during the data acquisition? Any information that could help assess whether there are mixed analyses that include alteration silicates, particularly at the rim? For instance, the EBSD maps in Fig. 4 show the laser pits and what looks like a bit incorporation of material outside the spinel grain in one of the spots (4b). The central spot in 4a completely incorporates a magnetite filled crack, and coprecipitated silicates, because the magnetite is not a source for Rb, Cs either. With a laser spot integrating over a large volume, avoiding altered bits that have an extremely high concentration compared to the near-zero (I expect ppb-range for these elements of interest, perhaps near-ppm for Li) is very hard, even in less altered samples. Si was not detected by APT, but the sample volume is much smaller.

(b) Instead of using Glitter, I would look at the raw data and look for spikes in the individual sweeps, in order to assess whether tiny inclusions vs a steady continuous signal are responsible for the observed FME enrichments.

(3) There is no critical assessment of the trace element data quality and whether these results make any sense. No comparison with known spinel partition coefficients. Rb and Cs are among the most incompatible lithophile trace elements. They should be near zero in any mantle phase, and the lowest in spinel. There is a good reason that there are few data published on these elements. These are difficult measurements.

(4) What is the point of the Atom Probe data? Using this advanced analytical marvel to demonstrate major element homogeneity is an extraordinary waste of resources. Is it possible to reprocess the raw data too look for enrichment and the distribution of the fluid-mobile elements I've been so skeptical about? If there are melt-derived tiny inclusions, or an alteration-related overprint along a microfracture, I would imagine that the APT should be able to reveal these.

[B] Origin and length scales of spinel heterogeneity

The arguments supporting the filtering methods that are applied here are flawed and severely biased, as I will show below. The length scales at which different and completely unrelated processes contribute to the perceived signal of heterogeneity are not recognized by the authors, and overextrapolated from their observation of the heterogeneous ANS peridotite.

(1) Dredge scale:

It's been well established that the uppermost mantle is highly heterogeneous. Even at the scale of a single dredge haul, major, trace, and isotopic heterogeneities in exhumed mantle peridotites have been described in great detail, focusing only on nominally residual peridotites (that means peridotites devoid of plagioclase or crosscutting gabbroic veins, or any other chemical sign of heterogeneity, such as a sample-scale compositional gradient). While the interpretation of various

extents of pre-existing depletion in the mantle is still debated in the community, it is obvious that a late-stage metasomatic overprint alone cannot be responsible for these isotopic heterogeneities.

(2) Sample scale:

- The V3306-IN18 sample studied by Hamlyn and Bonatti (1980) contains plagioclase. Like almost any plag-bearing peridotite from the ocean floor, the spinel is highly heterogeneous. HB80 pointed out and discussed this zoning in their pioneering article. Anybody who ever worked on plagioclase-bearing peridotites knows this.

- The 9604-4 sample from Brunelli et al (2003) is not residual but also contains plag.

- Several of the 15.20FZ peridotites, as well as Hess Deep and MAR 43N are truly heterogeneous on a sample scale. What they have in common is their highly refractory nature. I've seen something similar in some (not all) of the Gakkel Ridge harzburgites. That is something that requires further study. Perhaps the heterogeneity in the ANS peridotite is of similar origin, but a more detailed comparative study that includes new data acquisition is required to shed new light on this. Different cooling rates contribute to obliterating grain scale heterogeneities, imposed by whatever melt-fluid-peridotite interaction was responsible for this.

(3) Grain scale:

- KN180-14-44 (Dick et al 2010) does indeed contain one core measurement with 36.8% Al₂O₃; all 5 other measurements are rims and (small) interstitial grains, which have 38.9-40.3% Al₂O₃. That single core measurement does have low totals. This is not a good example.

- I agree, two of the spinel rims of L2627-04B (Cipriani et al 2009 EPSL) have high Al and a low Al core, but the same sample also has a high-Al rim, that is not mentioned in this section. This is a biased representation of the data.

- The other example from the same Cipriani study is not residual, but yes, both rim analyses appear to have higher Al. However, stating that "core showing low Al = 27.8 wt%" without mentioning that the other 4 core measurements have Al₂O₃ 30.1-32.2 wt% is bordering on scientific malpractice. You cannot just pick out the single measurement that suits your story.

So, stating that ~85% of all MOR-peridotite datasets have heterogeneous/modified spinels is incorrect and not supported at all by the published data. In fact, most of those peridotites are homogeneous, taking into account that there analytical uncertainties in the range of 1% (typical intralab precision; interlab accuracy is much worse), and other factors such as cooling-related exchange with pyroxenes (e.g. see Seyler et al., 2003, G3) that contribute to heterogeneity than the ANS-type high-Al-rim zoning. Whether that local-scale heterogeneity occurred recently, just prior to exposure on the ocean floor, is yet another matter. Some of it is, some could indeed result from a recent metasomatic overprint. In any event, the spinel data alone will not be sufficient to address this.

[C] Invalid conclusions and implications

I think the following conclusions are invalid:

L358-359: "In addition, we identify Cr-spinel as a carrier of fluid-mobile elements especially in subduction zone environments". Cr-spinel is not a carrier of fluid-mobile elements. These elements are unlikely to be bound in the spinel structure. Perhaps they can be present in mineral inclusions in some dunite-hosted chromites, but in the context of the data presented here, they more likely derived from low-temperature alteration along fractures measured by LA-ICPMS.

L361-362. "Furthermore, the heterogeneity of Cr# in Cr-spinel can instead be used as a tracer for Cr-spinel metasomatic modification". Not really. Within-sample and within-grain heterogeneity of Cr# in spinel is an indicator of disequilibrium. Metasomatic modification is one mechanism to do this. More important is the preservation of the heterogeneity by fast cooling and near-surface exposure, allowing researchers to study the origin of the heterogeneity.

L366-368. "In addition, Cr-spinel as a FME-carrying mineral may also provide a feeding mechanism for FME into the deep mantle chemical cycle because it breaks down at >1000 km depth". Not only are there no FME in spinel, the breakdown of spinel does not occur at depths of greater than 1000km. The transition of spinel peridotite to garnet peridotite occurs at a depth of less than 100 km.

Other statements in the conclusions are not as novel as the authors would like the reader to believe. For instance:

L349-351. "Our new analyses and a careful review of published results on Cr-spinel argue against the widely accepted concept that Cr-spinel is a reliable petrogenetic indicator for mantle melt generation and related tectonic environments". Heterogeneous spinel has been a reliable petrogenetic indicator of disequilibrium, and typically the first quantitative sign that the peridotite host is not a simple residue of partial melting.

[D] Minor quibbles

L1. I do not understand the word "melts" in the title. Maybe the term "residues" is more suitable, even though many of them are not strictly residual.

L70. Any modern (post-2000) EPMA or SEM has a standard BSE detector capable of resolving 0.1Z at high contrast, which corresponds to the change of about a change of Cr# 50 to 52. That is, if you're looking for variation. Larger variations in Cr#, such as observed in the studied sample, are impossible to miss, which is in my experience why routine BSE imaging is the first indicator when suspecting the presence of plagioclase in extremely serpentinized peridotites.

L212. “challenged much”. The authors are trying to overstretch the importance of their findings. There have been abundant studies pointing out high-temperature chemical overprints after partial melting. Importantly, retaining these modified peridotites at high temperatures in the mantle, even for short timescales, will quickly homogenize the spinel grains, even if the original overprint initially created zoned spinels. Other chemical signatures, such as LREE enrichments or decoupling of isotopically inferred ages, would still be preserved, while the spinels seem homogeneous and boring.

L224. “original igneous melts”. Again, the use of the word ‘melts’ I find very puzzling

L237. Typos. It’s Izu-Bonin-Mariana. Also in other places in the text (L427).

L343-345. This is not new. When Dick and Bullen (1984) was published, there were very few datapoints. For more than 25 years, the mantle peridotite community has learned and moved on from this geotectonic indicator, and nobody would believe such a simplistic concept. it's too late to make a big deal about a non-issue.

L368. The transition from a spinel peridotite to a garnet peridotite occurs at depths less than 100 km, and the redistribution of any spinel-hosted trace elements would occur there too.

L373. Too many significant digits

L374. Backscattered

L458. Please add the peak counting times.

L461-462. Si and Na are not reported elements in the supplementary tables, although it would have been useful to see them included in the data tables, in order to assess potential mixed analyses issues I have claimed to be an issue for the FME acquired by LA-ICPMS.

Point-by-point response to reviewer's comments

(Original reviewer's comments are in black, and our responses are in blue. Line numbers refer to line numbers in the revised clean version.)

Reviewer #1 (Remarks to the Author):

Review of the manuscript : “Is Cr-spinel a reliable petrogenetic indicator for mantle melts ? “ By Hamed Gamal El Dien et al.

I reviewed a previous version of this study and my impression was globally positive in spite of a few weaknesses. The present version has been significantly improved.

. New types of analyses (EBSD and Atom Probe Tomography) have been performed. These new data increase the strength of the authors' arguments in favour of their working hypothesis: Cr-spinel present in their samples from the Arabian-Nubian shield has clearly suffered partial re-equilibration with metasomatic melts/fluids.

. The lattice misorientation observed in a few grains allows the authors to discard solid-state plastic deformation as a - frequently invoked - modification mechanism of Cr-spinel composition.

. APT data contribute to further document the reverse zoning in Cr-spinel composition that was revealed through more classical analytical methods (EPMA, LA-ICP-MS,...).

. The microscopic mechanisms envisioned for the modification process (essentially the combined effects of corrosion and diffusion) are now explicitly addressed.

. The potential bias related to geometrical effects (i.e. 2-D sections to analyse compositional variations in a 3-D mineral) is now discussed.

. The section devoted to a re-evaluation of published data on Cr-spinel was also highly improved as the authors detail the way they have handled this huge amount of data. This exercise was far from being trivial given the heterogeneous nature of the published information, e.g. the sampling and analytical strategies and methods, the amount and quality of the meta-data associated to the chemical analyses, as the field and petrographic descriptions, the part (core vs. rim) of the crystal that was analysed, etc... This synthesis is impressive and definitely contributes to broaden the interest of the manuscript. The authors demonstrate that mantle peridotite are, in most occurrences worldwide, affected by melt/fluid/rock interaction (“metasomatism”) post-dating the decompression melting event.

. A better tribute is made to previous advocates of mantle metasomatism (although a reference to Kaczmarek and Müntener (2008)(*) is still lacking. These authors documented reverse zoning in Cr-Al in Cr-spinel and discussed its significance in the frame of melt percolation in peridotite.

. The minor problems have been addressed.

In conclusion, this is a great piece of work that will likely be highly cited by the students of mantle processes. My opinion is that this work is now ready for publication.

(*) Kaczmarek M.-A. and Müntener O. Juxtaposition of melt impregnation and high temperature shear zones in the upper mantle; field and petrological constraints from the Lanzo peridotite (Northern Italy). *Journal of Petrology*, 49, 2187-2220, 2008.

Georges Ceuleneer

Reply: Thanks for these comments that summarize our revisions following the reviewers' original comments, and the points on the importance of the present work to mantle petrology. The recommended reference has now been added to the manuscript.

Reviewer #2 (Remarks to the Author):

This revised paper presents compelling data to show that Cr-spinel grains are susceptible to cryptic

metasomatism that makes their utility as a melt-indicator at worst ineffective and at best questionable. The authors present rigorous characterization of a set of samples from an ancient ophiolite from the Arabian-Nubian Shield to show the relationship between Al-rich rims and FME, showing the effects of fluid-rock interactions with metasomatic fluids, likely in a fore-arc (FA) environment. They also use published data from MOR, FA and Back Arc environments to show that spinel heterogeneity is present in all these settings, and that significant overlap exists between the settings in terms of their Cr# - making the Cr# relatively ineffective as a tectonic indicator.

I believe this paper makes a worthy contribution that will have a large impact on the field and generate a lot of discussion and review of past studies. The authors have done significant work to improve the manuscript from the first submission. I have three remaining issues that need to be addressed before publication:

Reply: Thanks for these comments and the appreciation of our work.

1. There are several spots in the paper where the authors make statements that imply spinel Cr# are suspect – and it is not clear to me whether they mean this as a blanket statement or in a more nuanced way. If it is their intention to say that all Cr# interpretations are suspect (as it comes across to me as I read it, e.g., Line 36), then it's my opinion that this statement is too strong for the data they present. They have really good evidence for their locality and a good case for fore-arc or subduction related peridotite. They can suggest and hypothesize that the Cr# can be an issue for MOR peridotites based on compilation data (which they do), but there remains the possibility that in some cases the spinel do not violate the assumptions in the mantle-melting models and can still be used as a good indicator. I think the main take-away from this work is that there is strong evidence for cryptic metasomatism that requires in-depth characterization to detect. And only once homogeneity is proven can the spinel be used for mantle melting indices. Many historical studies may have not done this, but it does not mean they are invalid.

It also is written in a way that suggests that almost no one has worried about spinel heterogeneity before – e.g., in the abstract it says that almost all mantle petrologists have used this as a tool assuming a homogeneous, unaltered spinel. However, at several points through the paper they reference studies that have identified and discussed spinel zonation. It's possible that no previous studies have taken spinel zonation seriously enough, or interpreted it correctly in the view of the these authors, and that should be clarified.

Overall, both of these comments refer to the tone of the paper, which comes across too strongly in some spots and can be readily addressed by rephrasing and relatively minor editing.

Reply: Thanks for these two points. We have modified/rephrased these sentences in the abstract to make them more accurate. See lines 29-35.

We agree that some statements were a bit abrupt and we have rephrased them throughout the manuscript. Indeed, once a Cr-spinel has been proven to be homogeneous, it is valid to assume that the Cr# might provide information about the magmatic history of the rocks. See lines 342-349.

2. I like the revision of the conclusions and implications. I would like them to add a sentence or two that states what future studies will need to do to properly characterize spinel grains based on their findings. (e.g., BSE-EDS mapping, microprobe, # of spots, etc).

Reply: Thanks for this point. We have added some recommendations to the manuscript for future work on Cr-spinel chemistry. See lines 342-349.

3. Lastly, there are a few spots in the text that require some clarification. I've noted them in the Line Comments below:

Line 36: "making Cr # ineffective as a geotectonic and mantle melting indicator" I am not sure whether the authors mean to imply this is true for all Cr# measurements, or only for those in spinel that experienced metasomatism as evidenced by the reverse Cr-Al zonation. If the former, then I do not think they fully proved this and should rephrase the sentence in a way that is not so definitive. If the latter, then 'in these cases' should be added to this clause to clarify that the authors are referring to the reversed zoned spinel and not all spinel Cr#s.

Reply: The sentence has been modified as recommended. See line 35.

Line 68: "such reverse zoning is undetectable under microscope" – The way this is phrased implies that the other types of zoning are detectable. Certainly, in some cases (e.g., ferrite chromite rims), but often it's not detectable. Clarify the sentence.

Reply: The sentence has been clarified. See line 62.

Line 75: "describe a new type of Cr-spinel reverse zoning" – It's still unclear to me that this reverse zoning is "new" compared to the reverse zoning in the previous paragraph. Is the 'new' part the process that caused it? Because it sounds like it's the same type of reverse zoning – Al/Mg rich rim, Fe/Cr rich core? If it is the process that makes it new, then to avoid confusion, the 'new' should be added in the next sentence which describes the process.

Reply: The sentence has been modified. See lines 69-76.

Line 202: This is a big one - They state that Fe and Mg content of the cores and rims analyzed by atom probe are relatively constant when compared to the variation in Al and Cr. However, looking at Fig 5, all four elements vary by <1 to ~2 at%, so this presentation of the data and the statement are not sufficiently convincing. It seems that there is major zonation recorded in Fig 1 (by microprobe?). What is the explanation for the discrepancy in the magnitude of difference between the core and rim concentrations measured in wt% presented in Fig 1 and atom% presented in Fig 5?

Reply: We agree that there is a degree of variability in the Fe and Mg content between the core and the rim in the atom probe data. We have therefore removed the sentence "Whereas the Mg and Fe content is relatively constant between the core and rim (Fig. 5 and Table S2)".

In order to explain the compositional discrepancy between EPMA and atom probe, we have added the following text to the method section (lines 501-507): "The combination of multiple ionic species for each element and large thermal peak tails leads to difficulties in quantifying the composition of the analysed nano-scale domains. The estimation of the O composition from atom probe data is also renowned to be difficult (e.g., Gault et al., 2016). These limitations indicate that the composition calculated from the atom probe data will differ from expected stoichiometry and other techniques that apply standardization protocols (i.e. EPMA, LA-ICPMS...). However, the method used for the calculation of the composition in this study was consistent across all datasets, indicating that the differences between atom probe specimens are reliable."

We also modified the main text as follow (lines 195-198): “Although the major element composition calculated from the APT data differs from EMPA and LA-ICPMS data due to the lack of standardization protocols, the APT results confirm the Al enrichment and Cr depletion of the rim compared to the core (Fig. 5).”

Line 218: “would point to a transition of forearc/MOR setting”, this phrasing is confusing.

Reply: The sentence has been modified. See line 212.

Line 226: “should not use Cr-spinel data... in such peridotite” – What is meant by “such peridotite”? Any peridotite that has zoned spinel, or in particular ophiolitic peridotites that have an unclear origin?

Reply: “such peridotite” here refers to peridotites that had been affected by post melting complex history of melt/fluid-rock interactions and contain modified Cr-spinel as in our case. The sentence has been modified. See lines 216-218.

Line 363: “Our approach will also allow...” – What ‘approach’ that will allow mantle petrologists to use non-traditional isotopes are you referring to? This paper does not so much present an approach as it presents data to demonstrate the heterogeneity common to spinel grains. Please clarify the sentence.

Reply: The sentence has been modified. See lines 339-341.

Reviewer #3 (Remarks to the Author):

The main point of this manuscript is to demonstrate that Cr-spinels in mantle peridotites can be modified by post-melting processes, leading to an Al-Cr reverse zoning. According to the authors, this is a widespread process, and the post-melting modification distorts the inferred degree of partial melting commonly obtained from only measuring the spinel cores.

I disagree with this inference, and think the manuscript is flawed on several levels. There are issues with the data quality obtained on the sample presented in this study. In addition, the filtering process the authors use to emphasize the global importance of spinel heterogeneity is biased, not taking into account analytical errors of published data, not considering modern concepts of the origin of length scales of heterogeneity in the upper mantle, and a general overextrapolation when information is absent. For these reasons alone, I recommend waiting with the publication of these findings until all issues are resolved. The conclusions and implications are invalid.

The reverse zoning of Cr-spinel is actually an interesting observation, often seen in plagioclase peridotites, but it does not merit publication in this journal, even after extensive revisions and additional analyses. Perhaps the authors are on to something interesting, as there are indeed a couple of examples from the literature that have reverse spinel zoning. However, these are examples of non-residual peridotites, for which estimates of their degree of melting would be pointless to begin with, so the scope of the manuscript would have to change into a more positive message. In its current form, the underlying tone is too negative and passive aggressive, trying to persuade the reader that there’s a global problem resulting from peridotite researchers not working carefully.

Reply: We agree that we need to change the tone, which we did in this revised version. But we disagree that Al-Cr reverse zoning/heterogeneity of Cr-spinel is found in Pl-peridotites only, because it can also be found in residual peridotites. We provided 343 examples of

modified Cr-spinel at both sample and grain (within the grain core, or core to rim) scales. These include 272 examples of residual peridotites (not affected by any melt-rock interaction process) and 71 examples of non-residual peridotites (including dunite, plagioclase peridotite, gabbroic-pyroxenite veined samples, and metasomatized peridotite). More details can be found in Tables S1 and S3.

[A] Data quality

(1) Comparing the Cr-spinel data obtained by EPMA and LA-ICP-MS for Mn, Ti and Ni shows that there is a systematic disagreement between the datasets. The laser data always yield much higher concentrations than the more reasonable looking EPMA data.

[see plots in supplement]

Reply: Ti measured by EMPA here is usually below the detection limit (< 0.05 wt %) (Please see Morishita et al. (2018, American Mineralogist) for more details about detection limits for the Kanazawa electron microprobe) and this disagreement is the result of matrix effect. In addition, LA-ICPMS is more accurate in determining trace elements.

The LA-ICPMS data on the spinel were calibrated using NIST SRM 612, which is a silicate glass. Since the Cr-spinels have no Si, iron isotope ^{57}Fe was used as an internal reference mass. The problem is, in my opinion, that the low abundance of Fe (0.02% +/- 0.01% Fe_2O_3 according to the recommended values in Pearce et al 1997, used here. Instead, please refer to Jochum et al 2011, who report 51 +/-2 ppm Fe) in the NIST standard glass introduces an enormous uncertainty in the absolute trace element concentration of the measured spinels, and the data cannot be used at all. Did the measurements perhaps include Al? If so, Al may be used as an internal reference mass in the offline data reduction, and the data may still be salvaged. A troublesome byproduct of using ^{57}Fe as reference mass is the incorporation of magnetite veins by the big laser beam.

Reply: Fe^{57} instead of Al^{27} is a reasonable internal standard for measuring the trace element composition of spinel by using LA-ICPMS, because Al_2O_3 contents of spinels are over 50 wt% but the natural abundance of Al^{27} is 100% (this means that the detector of the Mass Spectrometer was saturated by the high counts from Al^{27}). Even if we did measure Al^{27} , the results would be associated with large counting errors. In addition, we used NIST612 as an external standard, because it is a well-tested external standard despite its low Fe content. Accordingly, our LA-ICPMS data has a reasonable accuracy.

(2) Ignoring the trace element accuracy for a second, the fluid-mobile elements Li, Sr, Rb, Cs appear to be enriched in the Al-rich domains, which dominantly occur at the spinel rims (Fig. 3). At first glance, this looks like a novel and fundamentally important observation. Unfortunately, any careful peridotite analyst knows that these data are not likely a high-temperature signature, but instead artefacts of mixed analyses of minute domains altered at low-temperature, incorporated by the large laser beam. I'd love to be corrected by the authors, but with the results and images presented here, I'm going to be conservative, requiring more rigorous evidence to be convinced that this is indeed a real high-temperature signal.

(a) Was Si acquired by LA-ICPMS? Were there any Si- or Fe-spikes during the data acquisition? Any information that could help assess whether there are mixed analyses that include alteration silicates, particularly at the rim? For instance, the EBSD maps in Fig. 4 show the laser pits and what looks like a bit incorporation of material outside the spinel grain in one of the spots (4b). The central spot in 4a completely incorporates a magnetite filled crack, and coprecipitated silicates, because the magnetite is not a source for Rb, Cs either. With a laser spot integrating over a large volume, avoiding altered bits that have an extremely high concentration compared to the near-zero (I expect ppb-range for these

elements of interest, perhaps near-ppm for Li) is very hard, even in less altered samples. Si was not detected by APT, but the sample volume is much smaller.

(b) Instead of using Glitter, I would look at the raw data and look for spikes in the individual sweeps, in order to assess whether tiny inclusions vs a steady continuous signal are responsible for the observed FME enrichments.

Reply: we agree that supplementary materials for the LA-ICPMS data are needed, and have added these to a new supplementary Figure S12. Moreover, we re-evaluated our LA-ICPMS signals to make sure there are no magmatic/fluid inclusions, silicate phases or secondary magnetite that could disturb the signals. The peak areas used for the calculation of trace elements do not include any spike signals for Si and Fe, which means that the fluid mobile elements data obtained by LA-ICPMS represents a Cr-spinel composition that is unrelated to low temperature phases such as serpentine or magnetite. Our results agree with X-ray mapping, electron microprobe (EMPA) (which show that Si (<0.03 wt %) and Na (<0.02 wt %) are below detection limits) and atom probe data, none of which detected Si.

(3) There is no critical assessment of the trace element data quality and whether these results make any sense. No comparison with known spinel partition coefficients. Rb and Cs are among the most incompatible lithophile trace elements. They should be near zero in any mantle phase, and the lowest in spinel. There is a good reason that there are few data published on these elements. These are difficult measurements.

Reply: please see the reply above and new supplementary Figure S12.

(4) What is the point of the Atom Probe data? Using this advanced analytical marvel to demonstrate major element homogeneity is an extraordinary waste of resources. Is it possible to reprocess the raw data to look for enrichment and the distribution of the fluid-mobile elements I've been so skeptical about? If there are melt-derived tiny inclusions, or an alteration-related overprint along a microfracture, I would imagine that the APT should be able to reveal these.

Reply: The atom probe data is critical to this study. Although the LA-ICPMS data did not recognise the occurrence of micrometre-scale silicate or Fe-oxide inclusions, nanometre-scale inclusions could be present, as seen in monazite for example (Fougerouse et al 2018; Seydoux-Guillaume et al 2019). The detection limit of atom probe for the fluid-mobile elements in spinel is unfortunately too high to be measured. However, the homogeneous chemical composition of major elements, including Fe, indicates that no Fe-oxide nano-scale inclusions or alteration features are present in the volume analysed. We have also reprocessed the data to check whether nano-scale silicate inclusions can be observed, but this yielded a null result. We have modified the text (lines 199-203) to better argue this point and clarify our approach.

Therefore, the combination of LA-ICPMS and atom probe tomography indicates that the trace elements in the rim of the spinel are not segregated in low temperature alteration phases, but are instead hosted in the spinel crystal structure as a solid-solution. This is a major finding of our study.

[B] Origin and length scales of spinel heterogeneity

The arguments supporting the filtering methods that are applied here are flawed and severely biased, as I will show below. The length scales at which different and completely unrelated processes contribute to the perceived signal of heterogeneity are not recognized by the authors, and overextrapolated from their observation of the heterogeneous ANS peridotite.

(1) Dredge scale:

It's been well established that the uppermost mantle is highly heterogeneous. Even at the scale of a single dredge haul, major, trace, and isotopic heterogeneities in exhumed mantle peridotites have been described in great detail, focusing only on nominally residual peridotites (that means peridotites devoid of plagioclase or crosscutting gabbroic veins, or any other chemical sign of heterogeneity, such as a sample-scale compositional gradient). While the interpretation of various extents of pre-existing depletion in the mantle is still debated in the community, it is obvious that a late-stage metasomatic overprint alone cannot be responsible for these isotopic heterogeneities.

Reply: We agree with this comment, which supports our statement that post-melting metasomatic processes can also affect the composition of the Cr-spinel. Also, our re-interpretation of high Al-Cr heterogeneity and lower Cr/Al ratio in Cr-spinel from MOR-peridotites to be of melt-rock interaction/metasomatism origin is supported by new geochemical modelling and observations of Brunelli et al. (2018, Nature Geosciences) from Vema at MAR, where peridotites affected by melt/rock interactions have lower Cr# than less affected ones.

(2) Sample scale:

- The V3306-IN18 sample studied by Hamlyn and Bonatti (1980) contains plagioclase. Like almost any plag-bearing peridotite from the ocean floor, the spinel is highly heterogeneous. HB80 pointed out and discussed this zoning in their pioneering article. Anybody who ever worked on plagioclase-bearing peridotites knows this.

Reply: This also supports our claim that peridotites affected by later stage interactions have highly heterogeneous Cr-spinel, such as in this sample that has inter-grain variations in Cr# spanning the range of 0.2 to 0.4 (almost the entire range for CIR peridotites). In addition, as we mentioned above, this heterogeneity is not only found in non-residual peridotites but also in residual ones. Please see detailed examples in Table S1.

- The 9604-4 sample from Brunelli et al (2003) is not residual but also contains plag.

Reply: We disagree here. The sample mentioned above doesn't contain plagioclase according to the authors' model composition and petrographic observations. Please see Table 2 of Brunelli et al. (2003).

- Several of the 15.20FZ peridotites, as well as Hess Deep and MAR 43N are truly heterogeneous on a sample scale. What they have in common is their highly refractory nature. I've seen something similar in a some (not all) of the Gakkel Ridge harzburgites. That is something that requires further study. Perhaps the heterogeneity in the ANS peridotite is of similar origin, but a more detailed comparative study that includes new data acquisition is required to shed new light on this. Different cooling rates contribute to obliterating grain scale heterogeneities, imposed by whatever melt-fluid-peridotite interaction was responsible for this.

Reply: We agree that the heterogeneous samples will have modified Cr-spinel. Please see Table S1 for more details about the modified Cr-spinel found in both residual and non-residual peridotites.

(3) Grain scale:

- KN180-14-44 (Dick et al 2010) does indeed contain one core measurement with 36.8% Al₂O₃; all 5 other measurements are rims and (small) interstitial grains, which have 38.9-40.3% Al₂O₃. That single core measurement does have low totals. This is not a good example.

Reply: Dick et al (2010) published a Cr-spinel dataset from Kane Megamullion (MAR), typically with data of one grain only per samples except for sample Kn180-2-4-2 that has two grains with a large range of Al content = 40.0 – 36.5 wt % and Cr# = 0.30 – 0.36 (Table S3). In addition, the above sample KN180-14-44 had only one core analysis and one rim analysis, with all other analyses cutting across core and rim. The results show high heterogeneity within one grain. For other examples of modified Cr-spinel by Dick et al (2010) see Table S1.

- I agree, two of the spinel rims of L2627-04B (Cipriani et al 2009 EPSL) have high Al and a low Al core, but the same sample also has a high-Al rim, that is not mentioned in this section. This is a biased representation of the data.

- The other example from the same Cipriani study is not residual, but yes, both rim analyses appear to have higher Al. However, stating that “core showing low Al = 27.8 wt%” without mentioning that the other 4 core measurements have Al₂O₃ 30.1-32.2 wt% is bordering on scientific malpractice. You cannot just pick out the single measurement that suits your story.

Reply: We do not think our dataset and main text show a biased view as we show a full set of examples in Table S1. We do not ignore that some samples have homogeneous Cr-spinel, however, what we are saying is that 85% of Cr-spinels are modified.

So, stating that ~85% of all MOR-peridotite datasets have heterogeneous/modified spinels is incorrect and not supported at all by the published data. In fact, most of those peridotites are homogeneous, taking into account that their analytical uncertainties in the range of 1% (typical intralab precision; interlab accuracy is much worse), and other factors such as cooling-related exchange with pyroxenes (e.g. see Seyler et al., 2003, G3) that contribute to heterogeneity than the ANS-type high-Al-rim zoning. Whether that local-scale heterogeneity occurred recently, just prior to exposure on the ocean floor, is yet another matter. Some of it is, some could indeed result from a recent metasomatic overprint. In any event, the spinel data alone will not be sufficient to address this.

Reply: We disagree with this statement. We carefully investigated the Cr-spinel in our comprehensive dataset and showed that modified Cr-spinel is a common feature in MOR-peridotites, including residual and non-residual ones. See Table S1 for all the examples at different scales.

[C] Invalid conclusions and implications

I think the following conclusions are invalid:

L358-359: “In addition, we identify Cr-spinel as a carrier of fluid-mobile elements especially in subduction zone environments”. Cr-spinel is not a carrier of fluid-mobile elements. These elements are unlikely to be bound in the spinel structure. Perhaps they can be present in mineral inclusions in some dunite-hosted chromites, but in the context of the data presented here, they more likely derived from low-temperature alteration along fractures measured by LA-ICPMS.

Reply: We disagree here. Our careful LA-ICPMS measurements and calculations of the data (data reduction) avoided any complex signal (see Fig. S12). In addition, APT and counting vs. time LA-ICPMS plots show that there are no inclusions measured, and the measured data represent that of the Cr-spinels.

L361-362. “Furthermore, the heterogeneity of Cr# in Cr-spinel can instead be used as a tracer for Cr-spinel metasomatic modification”. Not really. Within-sample and within-grain heterogeneity of Cr# in spinel is an indicator of disequilibrium. Metasomatic modification is one mechanism to do this. More important is the preservation of the heterogeneity by fast cooling and near-surface exposure, allowing researchers to study the origin of the heterogeneity.

Reply: Our study shows that the Cr-spinel were mostly modified by metasomatic processes.

L366-368. “In addition, Cr-spinel as a FME-carrying mineral may also provide a feeding mechanism for FME into the deep mantle chemical cycle because it breaks down at >1000 km depth”. Not only are there no FME in spinel, the breakdown of spinel does not occur at depths of greater than 1000km. The transition of spinel peridotite to garnet peridotite occurs at a depth of less than 100 km.

Reply: Our conclusions suggest the presence of FME in Cr-spinel. Please see the pioneering work of Ringwood on the stability of mantle minerals (for example Ringwood, 1962). In this work he said “Finally, between 900 and 1050 km, the spinel breaks down into denser, close packed phases, which persist to the core boundary at 2900 km. Two transformations are possible: (a) spinel → periclase + stishovite, or (b) spinel → periclase + MgSiO₃ (corundum structure).”

Other statements in the conclusions are not as novel as the authors would like the reader to believe. For instance:

L349-351. “Our new analyses and a careful review of published results on Cr-spinel argue against the widely accepted concept that Cr-spinel is a reliable petrogenetic indicator for mantle melt generation and related tectonic environments”. Heterogeneous spinel has been a reliable petrogenetic indicator of disequilibrium, and typically the first quantitative sign that the peridotite host is not a simple residue of partial melting.

Reply: We agree that the sentence quoted above is misleading and we clarified this point in the abstract and the last paragraph (lines 324-327). In addition, we suggest that Cr-spinel can be an excellent tracer for metasomatic processes (which means that we can still use it as petrogenetic indicator for post-melting processes).

[D] Minor quibbles

L1. I do not understand the word “melts” in the title. Maybe the term “residues” is more suitable, even though many of them are not strictly residual.

Reply: This word is now deleted from the title, and the title is modified to make it more positive.

L70. Any modern (post-2000) EPMA or SEM has a standard BSE detector capable of resolving 0.1Z at high contrast, which corresponds to the change of about a change of Cr# 50 to 52. That is, if you're looking for variation. Larger variations in Cr#, such as observed in the studied sample, are impossible to miss, which is in my experience why routine BSE imaging is the first indicator when suspecting the presence of plagioclase in extremely serpentinized peridotites.

Reply: Our BSE imaging of the studied samples, using various setups, was unable to detect the Al-Cr zoning in our samples and it is only observed by X-ray mapping. We did use a modern, high-end SEM (Tescan MIRA3). Our recommendation for future work on Cr-spinel is to first carry out systematic microanalyses and mapping to show if there is any Al-Cr heterogeneity in the Cr-spinel.

We do not see the point of comparing BSE signal for Cr-spinel and Plagioclase. Cr-spinel will always appear very bright, considering that spinel are Fe and Cr rich. Moreover, we do not understand the parallel with plagioclase. Our samples do not have any plagioclase. We emphasise that this feature is not found in Pl-peridotites only (please see Table S1).

L212. “challenged much”. The authors are trying to overstretch the importance of their findings. There have been abundant studies pointing out high-temperature chemical overprints after partial melting. Importantly, retaining these modified peridotites at high temperatures in the mantle, even for short timescales, will quickly homogenize the spinel grains, even if the original overprint initially created zoned spinels. Other chemical signatures, such as LREE enrichments or decoupling of isotopically inferred ages, would still be preserved, while the spinels seem homogeneous and boring.

Reply: This sentence has been modified. Lines 204-205.

L224. “original igneous melts”. Again, the use of the word ‘melts’ I find very puzzling

Reply: This sentence has been deleted.

L237. Typos. It’s Izu-Bonin-Mariana. Also in other places in the text (L427).

Reply: Modified.

L343-345. This is not new. When Dick and Bullen (1984) was published, there were very few datapoints. For more than 25 years, the mantle peridotite community has learned and moved on from this geotectonic indicator, and nobody would believe such a simplistic concept. it's too late to make a big deal about a non-issue.

Reply: We disagree with this statement since the Cr# of Cr-spinel is still widely used in the literature as a geotectonic indicator.

L368. The transition from a spinel peridotite to a garnet peridotite occurs at depths less than 100 km, and the redistribution of any spinel-hosted trace elements would occur there too.

Reply: please see the reply above.

L373. Too many significant digits

L374. Backscattered

L458. Please add the peak counting times.

Reply: The LA-ICPMS ablation profiles for counting vs time are available in the supplementary information (Fig. S12).

L461-462. Si and Na are not reported elements in the supplementary tables, although it would have been useful to see them included in the data tables, in order to assess potential mixed analyses issues I have claimed to be an issue for the FME acquired by LA-ICPMS.

Reply: Si and Na are not reported in Table S2 because they are below the detection limits (<0.03 and <0.02, respectively). Please see Morishita et al. (2018, American Mineralogist) for more details about detection limits for the Kanazawa electron microprobe. See also Fig. S12 for LAICP peaks (Si and Fe) that do not show any presence of silicates and magnetite inclusions.

REVIEWERS' COMMENTS:

Reviewer #2 (Remarks to the Author):

Thank you for your responses to my review. All of my concerns were addressed.

Reviewer #4 (Remarks to the Author):

I carefully read the paper "Is Cr-spinel a reliable petrogenetic indicator?" by El Dien et al. as well as the letter with the answers to the previous reviews. It appears to me that the authors did an excellent job for this new version. In particular they took into account the comments from the previous reviewers in the best possible way. I have no more remarks and comments to do and I propose an acceptance in the present form. I only noticed few typo errors (see the attached file)

Yours sincerely

Michel Grégoire

Point-by-point response to reviewer's comments

(Original reviewer's comments are in black, and our responses are in blue. Line numbers refer to line numbers in the revised clean version.)

Reviewer #2 (Remarks to the Author):

Thank you for your responses to my review. All of my concerns were addressed.

Reply: Thanks for your constructive comments during the review process.

Reviewer #4 (Remarks to the Author):

I carefully read the paper "Is Cr-spinel a reliable petrogenetic indicator?" by El Dien et al. as well as the letter with the answers to the previous reviews. It appears to me that the authors did an excellent job for this new version. In particular they took into account the comments from the previous reviewers in the best possible way. I have no more remarks and comments to do and I propose an acceptance in the present form. I only noticed few typo errors (see the attached file)

Yours sincerely

Michel Grégoire

Reply: Many Thanks!. Typo errors has been addressed.